# Peroxisome Proliferator-Activated Receptors and Caloric Restriction—Common Pathways Affecting Metabolism, Health, and Longevity

**DOI:** 10.3390/cells9071708

**Published:** 2020-07-16

**Authors:** Kalina Duszka, András Gregor, Hervé Guillou, Jürgen König, Walter Wahli

**Affiliations:** 1Department of Nutritional Sciences, University of Vienna, 1090 Vienna, Austria; Andras.Gregor@univie.ac.at (A.G.); juergen.koenig@univie.ac.at (J.K.); 2Toxalim (Research Centre in Food Toxicology), INRAE, ENVT, INP-Purpan, UMR 1331, UPS, Université de Toulouse, F-31027 Toulouse, France; herve.guillou@inrae.fr; 3Lee Kong Chian School of Medicine, Nanyang Technological University Singapore, Singapore 308232, Singapore; 4Center for Integrative Genomics, University of Lausanne, Le Génopode, CH-1015 Lausanne, Switzerland

**Keywords:** nuclear receptors, PPARs, nutrition, caloric restriction

## Abstract

Caloric restriction (CR) is a traditional but scientifically verified approach to promoting health and increasing lifespan. CR exerts its effects through multiple molecular pathways that trigger major metabolic adaptations. It influences key nutrient and energy-sensing pathways including mammalian target of rapamycin, Sirtuin 1, AMP-activated protein kinase, and insulin signaling, ultimately resulting in reductions in basic metabolic rate, inflammation, and oxidative stress, as well as increased autophagy and mitochondrial efficiency. CR shares multiple overlapping pathways with peroxisome proliferator-activated receptors (PPARs), particularly in energy metabolism and inflammation. Consequently, several lines of evidence suggest that PPARs might be indispensable for beneficial outcomes related to CR. In this review, we present the available evidence for the interconnection between CR and PPARs, highlighting their shared pathways and analyzing their interaction. We also discuss the possible contributions of PPARs to the effects of CR on whole organism outcomes.

## 1. Introduction

Caloric restriction (CR) is one of the primary interventions for weight loss and health maintenance. As early as the 16th century, Luigi Cornaro (1484–1566) described the beneficial effects of this approach in his “Discorsa della vita sobria”. Later, at the beginning of the 20th century, the first experimental evidence emerged when Osborne et al. reported that CR slowed the growth of rats but prolonged their lifespan [1]. In rats, a CR of 40% applied from weaning onward has been linked to a lifespan extension of almost two-fold [2]. In fact, CR has been associated with increases in mean and maximum lifespan, regardless of sex, in multiple species, including various rat and mouse strains, yeasts, worms, fruit flies, fishes, hamsters, dogs, cows, and owls [3]. The effects of CR in these organisms include reduced neurodegenerative disease incidence, diminished rates of age-specific mortality, and a lower incidence of cancer, diabetes, atherosclerosis, and cardiovascular disease. CR also is linked to delayed onset of age-related processes, such as immunosenescence, sarcopenia, and atrophy of the brain gray matter [3,4,5,6,7]. In monkeys, CR leads to diabetes suppression and a reduced incidence of neoplasia and cardiovascular diseases by up to 50% [6]. These effects have been attributed to a reduction in major risk factors, including cholesterol, C-reactive protein, blood pressure, and intima-media thickness of the carotid arteries [7,8,9]. Beneficial outcomes of CR have been consistently reported, which supports this approach considering that distinct CR protocols are used in different publications. CR applied in diverse studies ranges from 10% to up to 50% of daily caloric intake. Furthermore, the length of CR varies from a few weeks to life-long treatment. Additionally, some protocols restrict all nutrients, whereas others limit macronutrients only and supplement micronutrients in order to investigate selectively the impact of calorie reduction and prevent malnutrition, making a distinction between “dietary restriction” and “energy restriction” [10]. As expected, the type of CR protocol influences the magnitude of outcomes [11,12]. Moreover, the results obtained from experimental models cannot be directly translated to humans [13]. Therefore, it is important to compile the results of multiple studies to identify common patterns of responses regardless of the type of CR. A comparison of the responses from different species may help to draw a more comprehensive picture of the outcomes of CR.

CR has been tied to a complex network of pathways implicating insulin-like growth factor 1 (IGF-1), sirtuins (SIRTs), adenosine monophosphate (AMP)-activated protein kinase (AMPK), and target of rapamycin (TOR). The sympathetic and neuroendocrine systems, as well as thyroid hormones, adipokines, and ghrelin, also have been associated with the beneficial outcomes of CR [4]. This ensemble of processes associated with CR affects the whole body, manifesting in reduced inflammation, body fat mass, resting metabolic rate, and body temperature and improved insulin sensitivity [14]. As a result of the variety of outcomes related to CR and the complexity of the contributing pathways, the exact mechanisms underlying these health benefits are still not well understood. However, the results of ongoing studies have filled in some parts of this puzzle. This review focuses on one important piece: the role of peroxisome proliferator-activated receptors (PPARs) in outcomes associated with CR.

## 2. PPARs

PPARs form a subfamily of the ligand-dependent nuclear receptor superfamily, which consists of three isotypes, PPARα (NR1C1), PPARβ/δ (NR1C2), and PPARγ (NR1C3), each coded by a separate gene [15,16,17]. PPARs play major roles in a broad spectrum of biological processes, including cell proliferation and differentiation, fatty acid (FA) and eicosanoid signaling, bone formation, tissue repair and remodeling, insulin sensitivity, and, in particular, glucose and lipid metabolism (Figure 1). They are transcription factors that form heterodimers with retinoid X receptors (RXRs) and bind to specific peroxisome proliferator response elements (PPREs) in the transcription regulatory region of their target genes. A number of coactivators and corepressors modulate PPAR activity, either stimulating or inhibiting receptor function [18]. Two important PPAR corepressors are the nuclear corepressor 1 (NCoR1) and the silencing mediator of retinoic acid and thyroid hormone receptor (SMRT) [19,20,21], which are recruited to PPARs in the absence of ligands and limit PPAR transcriptional activity [22,23]. Coactivators encompass various proteins, including factors with histone acetylase activity [cAMP response element-binding (CREB) protein (CBP)/p300 and steroid receptor coactivator 1–3 complexes], helicases [PPAR A–interacting complex (Pric)285, Pric320/chromodomain helicase DNA binding protein 9], and an ATPase in the SWItch/sucrose non-fermentable (SWI/SNF) complex, and non-enzymatic activators that can be found in the active PPAR transcriptional complex [PPARγ coactivator (PGC)-1α, PGC-δ/PGC-1–related estrogen receptor coactivator, mediator of RNA polymerase II transcription subunit/TRAP220/PPAR-binding protein, PPAR-interacting protein/nuclear receptor coactivator 6, SWI/SNF-related, matrix-associated, actin-dependent regulator of chromatin, subfamily D, member 1] [18]. The characteristic feature of the PPAR ligand-binding cavity is its 3-4-fold larger size compared to other nuclear receptors. Thus, PPARs can accommodate and bind a variety of natural and synthetic lipophilic acids [24,25,26,27]. Synthetic ligands of PPARs are widely used in clinical practice to treat glucose and lipid disorders and in the prevention and treatment of cardiovascular and metabolic diseases [28,29,30]. Synthetic ligands can be specific for each PPAR isotype or activate two (saroglitazar, elafibranor) [31,32] or all three (bezafibrate, lanifibranor) isotypes [33,34]. Natural ligands of PPARs include various FAs, phospholipids, prostaglandins, prostacyclins, and leukotrienes [35,36] linking the activity of PPARs to nutrition, metabolism, and inflammation. In addition to ligands, PPAR transcriptional activity can be modulated by post-translational modifications including phosphorylation, ubiquitination, O-GlcNAcylation, and SUMOylation [29,37,38,39,40,41,42,43].

### 2.1. PPARα

The first cloned PPAR, now known as PPARα, was initially identified as the molecular target of xenobiotics inducing hepatic peroxisome proliferation in rodents [44]. PPARα is particularly abundant in organs with high demand for FA catabolism, such as liver and brown adipose tissue (BAT) [45]. In these tissues, this transcription factor coordinates several aspects of metabolism by modulating the expression of genes involved in peroxisomal and mitochondrial β-oxidation, FA transport and catabolism, ketogenesis, and gluconeogenesis [35]. In line with this role of PPARα, the L162V *Pparα* gene polymorphism, which results in a PPARα variant with lower transcriptional activity, is associated with increased serum levels of triglycerides (TGs), total cholesterol, low-density lipoprotein cholesterol, and apolipoprotein (apo) A1 and apoB [46,47,48,49,50], influencing the onset and progress of type 2 diabetes (T2D) [51,52].

PPARα serves as a sensor of nutritional status. In starvation, when the organism mobilizes stored FAs, PPARα in hepatocytes increases FA uptake and β-oxidation and stimulates hepatokine gene expression [53,54,55,56]. Therefore, PPARα adapts the rates of FA catabolism and ketone body synthesis to energy depletion. Furthermore, in the mouse liver, PPARα is a key factor in the metabolic adaptation to sepsis. Deficiency in hepatic PPARα causes an impaired metabolic response, and upon bacterial infection, PPARα-null mice have a higher mortality rate. These observations suggest that activating PPARα may be considered a plausible metabolic intervention for improving sepsis outcomes. In support of this hypothesis, non-surviving critically ill patients have reduced PPARα activity in their livers [57].

The natural ligands of PPARα are FAs and their derivatives [58]. Synthetic fibrate PPARα agonists are generally used to treat patients with primary hypercholesterolemia, mixed dyslipidemia, and hypertriglyceridemia. Natural and synthetic PPARα agonists have been discussed extensively [29,30,59,60], and their effect appears to be based mostly on the stimulation of cellular lipid trafficking and β-oxidation [61,62]. However, the need persists for better drug candidates that selectively target PPARα without the side effects of fibrates. The recent development of selective PPARα modulators may fill this gap [63].

### 2.2. PPARβ/δ

PPARβ/δ is ubiquitously expressed, with particularly high levels in the skin, gastrointestinal tract, liver, kidney, and various parts of the central nervous system [64,65]. PPARβ/δ is involved in cell proliferation, differentiation, and survival, and it plays a role in tissue repair [66,67,68,69]. It is necessary for placental and gut development and is involved in the control of energy homeostasis [70,71,72,73,74]. Metabolically, PPARβ/δ promotes FA oxidation in adipose tissue and skeletal muscle, leading to improved lipidemia, and it stimulates overall energy expenditure, protecting against diet-induced obesity and insulin resistance [75,76,77,78]. It is probably best known for stimulating energy expenditure in muscles and increasing exercise capacity [79]. Under regular exercise, PPARβ/δ induces a switch to an increased amount of type I muscle fibers, enhancing mitochondrial function and fat oxidation and boosting endurance [80]. For this reason, this receptor also is a target for doping substances, so much so that the World Anti-Doping Agency added the PPARβ/δ ligands GW501516 and other related chemicals to its prohibited list in 2009. These synthetic ligands (GW501516, GW0742, and L-165041) selectively activate PPARβ/δ at very low concentrations and are often applied in both in vivo and in vitro research models [81], but no PPARβ/δ agonist has been used in clinical practice. Nevertheless, PPARβ/δ remains a potential therapeutic target in metabolic diseases. Other synthetic and natural PPARβ/δ agonists recently have been discussed elsewhere [29].

### 2.3. PPARγ

PPARγ is recognized primarily for its insulin-sensitizing properties and its role as a master regulator of adipogenesis [82,83,84]. It also has been identified as a promising therapeutic target for several types of cancer because it limits tumor progression by decreasing cell proliferation [85,86,87], increasing cell differentiation [85,88], inducing apoptosis [85,86,89,90,91,92,93], and inhibiting angiogenesis [94]. In addition, PPARγ plays a role in multiple processes including inflammation, adipogenesis, FA storage, and lipid and glucose metabolism [82,95,96,97,98,99]. We have described novel roles for intestinal PPARγ in long-chain FA processing in the intestinal epithelium [100] and in the regulation of body adiposity through sympathetic nervous system signaling [101], and we identified it as a major regulator of mucosal defenses upon high-fat diet feeding in mice [102]. However, its effects are not exclusively beneficial. For instance, it has recently been reported that PPARγ has a carcinogenic effect in advanced brain metastases [103] and liver cancer [104].

PPARγ occurs in two isoforms, PPARγ1 and PPARγ2, with the latter distinguished by 30 and 28 extra amino acids at the N-terminus in mouse and human, respectively [105]. PPARγ2 is found at high levels in adipose tissues [106,107], whereas PPARγ1 shows a broader expression pattern. In addition to adipose tissue, PPARγ1 is expressed in the gut, brain, and vascular cells and in specific types of immune and inflammatory cells [65,105,108,109,110]. A plethora of factors induce PPARγ expression and activity, and not all are ligands. They can include FAs and their metabolites [25,111], eicosanoids [25], prostaglandins [112], phytanic acid [113], and multiple nutrients, along with glutamine, curcumin, capsaicin, ginsenosides, and vitamin E, all of which have been reported to exhibit anti-inflammatory properties [114]. Of note, the presence of specific bacterial strains, their metabolites, and bacterial by-products [115,116,117,118,119,120] also stimulate PPARγ expression and/or activity.

Synthetic agonists of PPARγ can be divided into two groups: classical full agonists, which are represented by the thiazolidinediones (TZDs) [121] such as troglitazone, rosiglitazone, and pioglitazone, and partial agonists, which were developed to reduce the side effects of full agonists, including weight gain and bone loss [122,123]. PPARα/γ dual agonists exert positive influences on both lipid and glucose metabolism. They not only have antidiabetic capacity but also are hypolipemic, anti-inflammatory, hypotensive, and antiatherogenic and show anticoagulant action leading to improved endothelial function [30,31,59,124,125,126,127,128,129,130].

### 2.4. PPARs in CR

The effects of CR on PPAR expression vary among different tissues and organs. PPAR expression either remains unchanged [PPARα, -β/δ, and -γ in the heart, PPARα in white adipose tissue (WAT), PPARγ in the liver], decreases (PPARβ/δ in the liver, PPARα in the spleen, PPARα, -β/δ, and -γ in muscle), or increases (PPARα in the liver, heart, and intestinal epithelium) [131,132,133,134,135,136,137,138,139]. In rat kidneys, PPAR mRNA and protein levels and DNA-binding activity decrease with age, but a 60% CR blunts this reduction [140].

PPARs can contribute to CR-related outcomes by modulating multiple pathways connected with metabolism, insulin sensitivity, inflammation, and oxidative stress, all of which we describe at the molecular level in this review. Most of the published works on the impact of PPARs in CR concern PPARα. CR induces the upregulation of PPARα expression in the heart, where it increases lipid use [139]. In fact, 19% of the hepatic genes involved in lipid metabolism, inflammation, and cell growth and whose expression is changed by CR depend on PPARα, as shown by the absence of regulation of these processes in PPARα-null mice [141]. Therefore, ligands of PPARα have been proposed to serve as mimetics of CR [141]. Although less studied so far, the two other PPAR isotypes likewise modulate multiple pathways in relation to CR, as we discuss below.

## 3. Major Pathways Affected by CR

### 3.1. mTOR

The mammalian (m)TOR pathway is a major nutrient sensor signaling pathway known to regulate longevity (Figure 2). TOR is a well-conserved Ser/Thr protein kinase that belongs to the family of phosphatidylinositol 3 (PI3) kinase-related kinases [142,143]. It functions as an essential part of two complexes, mTORC1 and mTORC2, which have some proteins in common and some different proteins between them [142]. mTORC1 comprises the following core subunits: mTOR, mLST8 (mammalian lethal with sec-13 or GβL), DEPTOR (DEP domain-containing mTOR-interacting protein Tti1/Tel2 complex), PRAS40 (proline-rich Akt substrate of 40 kDa), and Raptor (regulatory-associated protein of mammalian target of rapamycin). mTORC2 is composed of mTOR, mLST8, DEPTOR, the Tti1/Tel2 complex, Rictor (rapamycin-insensitive companion of mTOR), mSin1 (mammalian stress-activated MAP kinase-interacting protein 1 or MAPKAP1), and protor1/2 (protein observed with Rictor 1 and 2) [144,145,146,147]. The configuration of each of these two complexes is conserved from yeast to mammals [148]. mTORC1 is sensitive to inhibition by rapamycin and plays essential roles in the regulation of mRNA translation and autophagy. Cellular energy and nutrient status regulate it directly, whereas mTORC2, which is not rapamycin sensitive, functions mainly as an important regulator of the cellular actin cytoskeleton [149,150].

Rheb (Ras homolog enriched in the brain) is a GTPase that in its GTP-bound form directly binds to and activates mTOR [151,152,153,154]. Rheb activity is inhibited by the heterodimer complex of tuberous sclerosis proteins 1 and 2 (TSC1 and TSC2) [155,156,157,158,159,160]. TSC1/2 mediates for mTORC1 many of the upstream signals from growth factors, such as insulin and IGF-1, which stimulate the PI3K and Ras pathways. The effector kinases of these pathways, including Akt (or protein kinase B or PKB), extracellular-signal-regulated kinase 1/2 (ERK1/2), and ribosomal protein S6 kinase β-1 (S6K1), directly phosphorylate and inactivate the TSC1/TSC2 complex, leading to the activation of mTORC1 [154,157,161,162,163,164,165,166,167]. Rheb also can transmit upstream signals from the p38β–PRAK pathway, which is activated upon glucose starvation [168]. Finally, as a core component of mTORC2, mTOR functions as a tyrosine protein kinase that promotes activation of the insulin receptor and IGF-1 receptors [169]. These interactions illustrate the tightly interconnected signaling between mTOR and insulin.

The mTOR pathway integrates inputs from major intracellular and extracellular physiological stimuli (growth factors, stress, energy balance, oxygen, amino acids) and controls many major downstream processes, including macromolecule synthesis, autophagy, cell cycle, growth, and metabolism [142,143,170]. For example, the canonical Wnt pathway, AMPK, some pro-inflammatory cytokines such as tumor necrosis factor-α (TNFα), and the hypoxia-inducible proteins REDD1 and REDD2 modulate mTORC1 activity via TSC1/2 [171,172,173,174,175,176]. In addition to phosphorylating TSC1/2, AMPK phosphorylates Raptor, leading to the allosteric inhibition of mTOR [177]. mTORC1 activity is further regulated by lipid-derived signaling molecules (phosphatidic acid) [178], the redox status of the cell [179], and amino acids, particularly leucine and arginine [180,181]. DNA damage also signals to mTORC1 through multiple mechanisms, all of which require p53-dependent transcription, induction of the expression of TSC2 and phosphatase and tensin homolog deleted on chromosome 10 (PTEN), and AMPK activation [182,183,184].

Downstream signaling of mTORC1 controls autophagy and energy metabolism, including the glycolytic flux, lipid synthesis [185,186,187,188], and cholesterol synthesis via the activation of sterol regulatory element-binding protein (SREBP) 1/2 [185,189,190]. mTORC1 also promotes anabolism in the fed state by controlling lipid metabolism in the liver through the modulation of *Srebp1c* expression, which is a regulator of lipogenesis and lipid storage [191,192].

Under mTORC1 regulation, mitochondrial DNA content and the expression of genes involved in oxidative metabolism increase. mTORC1 exerts this effect in part by mediating the nuclear association between PPARγ coactivator 1α (PGC-1α) and the transcription factor Yin-Yang 1, which positively regulates mitochondrial biogenesis and oxidative function [193] (see the section on mitochondrial function).

The activation of mTOR also leads to the phosphorylation of many target proteins related to the translational machinery and ribosome biogenesis, such as p70 ribosomal S6 kinase (S6K) and eukaryotic initiation factor 4E-binding protein (4E-BP) [170,194,195,196,197,198,199]. The regulation of protein metabolism also is a much-recognized function of mTOR. Amino acid activation of mTORC1 promotes protein synthesis via the activation of S6K and/or inhibition of 4E-BP, whereas the inactivation of mTORC1 promotes the degradation of damaged proteins and intracellular organelles via autophagy [200,201] (Figure 2).

mTORC2 functions mainly as an important regulator of the actin cytoskeleton through its stimulation of F-actin stress fibers, paxillin, RhoA, Rac1, Cdc42, and protein kinase C (PKC) α [146]. mTORC2 phosphorylates Akt [202,203] and thus affects metabolism and cell survival. mTORC2 also directly activates SGK1, which is a kinase controlling ion transport and growth [204]. Both Akt and SGK1 phosphorylate FoxO1/3a [205,206,207].

As a result of its role as an amino acid sensor, the TOR pathway has been proposed as a mediator of CR. The high activity of mTORC1 is a major driving force of aging, whereas the suppression of mTOR is tied to many of the benefits associated with CR, including lifespan extension [208,209,210,211], as has been demonstrated in yeast [208,212], worms [209], and flies [210]. Rapamycin treatment slightly extends the lifespan in flies subjected to CR [213]. In yeast, CR does not further extend the lifespan in the absence of *TOR1*, which is one of the two *TOR* genes in yeast, suggesting that TOR inhibition and CR promote lifespan via a common mechanism [208]. Similarly, in *C. elegans*, using RNA interference against TOR or autophagy genes in *eat-2 * mutant worms, which have impaired feeding behavior and are used as a genetic model for CR, does not extend the lifespan [214,215]. Furthermore, the inhibition of one of the principal targets of TOR signaling, S6K, extends the lifespan of *eat-2 C. elegans* [216]. Of note, mTOR activation in the rat’s brain results in reduced food intake by promoting the expression of the orexigenic neuropeptide Y and agouti-related peptide in the hypothalamus [217,218]. These data suggest that CR and TOR inhibition promote lifespan via overlapping pathways.

### 3.2. mTOR and PPARα

The various direct or indirect interactions between mTOR and PPARs have diverse effects on FA synthesis, glucose metabolism, oncogenesis, and immune system activation (Figure 3). First, mTORC1 regulates hepatic ketone body production in response to fasting [219]. mTORC1 activity is low during fasting, which correlates with increased PPARα activity. Consequently, mice with a constitutive activation of mTORC1 in the liver and with correspondingly low PPARα cannot induce ketogenesis when fasted [219]. As alluded to above, PPARα is particularly known for its role in fasting-triggered FA oxidation and lipid metabolism in the liver. The intertwined functions of mTORC1 and PPARα lie in the mTORC1 inhibition of PPARα activity during feeding and consequently blocking hepatic ketogenesis.

In the fed state, the insulin-dependent PI3K pathway activates mTORC1, promoting the cytoplasmic and nuclear localization of the PPARα corepressor NCoR1 and leading to the inhibition of PPARα activity and ketogenesis [219]. Fasting leads to decreased mTORC1 and consequently S6 kinase 2 activity, which promotes the cytoplasmic localization of NCoR1. In the absence of nuclear NCoR1, the increased transcriptional activity of PPARα enhances the FA oxidation that generates substrates for ketogenesis and gluconeogenesis [219,220]. In fact, PPARα also stimulates the expression of mitochondrial hydroxymethylglutaryl-CoA synthase (HMGCS), which is a rate-limiting enzyme of ketogenesis that catalyzes the condensation of acetyl coenzyme A (acetyl-CoA) and acetoacetyl-CoA to generate 3-hydroxy-3-methylglutaryl (HMG)-CoA and CoA [37]. The CR-triggered increase in the intestinal HMG-CoA synthase 2 (HMGCS2) expression alters the regeneration and differentiation capacity of intestinal steam cells. The result is a reduction in the differentiation markers of secretory cells with the promotion of enterocytes, changing the functionality of the intestine [221]. The subsequent refeeding of CR mice leads to reduced HMGCS2 levels and an increased expression of Paneth and goblet cell markers [221]. Additionally, in the intestine, the age-related increased activity of mTOR inhibits PPARα, resulting in higher levels of Notum and decreased Wnt signaling, consequently diminishing the regenerative function of stem cells in the Paneth cell niche [222]. Similarly, mTORC1 activity is elevated in the livers of old mice [219], which correlates with reduced PPARα activity and hepatic ketogenesis during aging [223,224,225]. mTORC1 inhibition is sufficient to prevent both the aging-induced loss of PPARα activity and ketone body production [219].

Another type of functional interaction between mTOR and PPARα relates to ligand production for the latter by FA synthase (FAS). In the fed state, mTORC1 mediates insulin-dependent phosphorylation and thus the inhibition of cytoplasmic FAS, limiting ligand generation. During fasting, when mTOR is inhibited, nonphosphorylated active cytoplasmic FAS promotes the synthesis of endogenous PPARα ligands [37].

In several organs, CR induces autophagy, which is a process that integrates mTOR and PPARα. To protect the liver from acute failure, PPARα-mediated induction of autophagy attenuates a lipopolysaccharide (LPS)-induced pro-inflammatory response [226]. Furthermore, agonists of PPARα (GW7647 and WY-14,643) regulate multiple genes involved in autophagy and lysosomal biogenesis and function, such as the transcription factor EB, which is a master gene for lysosomal biogenesis [227]. Of interest, a protein called farnesoid X receptor (FXR) is activated in the fed liver and suppresses autophagy. PPARα, activated in fasted and CR livers, regulates genomic circuits that are complementary to those under FXR control. Moreover, FXR stimulates the hepatic expression of PPARα [228]. These findings highlight how the liver senses nutrient status and how these two nuclear receptors translate this status in autophagy regulation [229,230].

### 3.3. mTOR and PPARβ/δ

Relatively little evidence connects mTOR and PPARβ/δ functions. In human lung carcinoma cells, nicotine activates PPARβ/δ expression through PI3K/mTOR [231], whereas the PPARβ/δ agonist GW501516 stimulates the growth of these cells through the inhibition of PTEN expression [232], indicating the interplay between the two pathways. Additionally, PPARβ/δ may modulate mTOR activity by mediating the metabolism of FAs and the production of phosphatidic acid, which is a metabolite that directly activates the mTOR complex by increasing its stability and activity. Phosphatidic acid responsiveness has been proposed as a lipid precursor sensing mechanism for the biosynthesis of cell membranes in the context of cell division and cell mass increase [233].

### 3.4. mTOR and PPARγ

As noted, PPARγ is a master regulator of adipogenesis. In parallel, mTORC1 senses growth factors and nutrients that drive adipose tissue accumulation. The inhibition of mTORC1 impairs adipogenesis and adipocyte maintenance in vitro [187,234,235,236,237], at least in part by modulating PPARγ expression and transcription [187,188,238,239]. mTORC1 may activate PPARγ through SREBP1, which promotes the production of endogenous PPARγ ligands [240,241]. Once activated by its natural or synthetic ligands, PPARγ stimulates mTORC1 and AMPK and upregulates TG-derived FA uptake, lipoprotein lipase activity, and accumulation in subcutaneous WAT and BAT. Chronic mTOR inhibition attenuates these processes, which leads to hyperlipidemia. These observations imply that mTOR regulates the hypolipidemic and lipogenic effects of PPARγ [239], as also suggested by the rapamycin inhibition of adipocyte differentiation [187,234,237]. Additionally, rapamycin reduces the phosphorylation of lipin-1 [242], which is a phosphatidic phosphatase that is involved in phospholipid and TG synthesis as well as the coactivation of many transcription factors linked to lipid metabolism, including PPARγ, PPARα, and PGC-1α [243,244,245]. A model has been proposed for nutrient and insulin signaling during adipogenesis in which the mTOR and PI3K/Akt pathways respectively act in parallel to regulate PPARγ activation [187], revealing pathway synergy in the context of metabolism.

Conversely, PPARγ also affects mTOR activity. It inhibits mTOR signaling with a protective action in rats with status epilepticus [246], and after activation with rosiglitazone, it stimulates mTORC1 and AMPK [239]. The interaction between PPARγ and mTOR has been particularly well demonstrated in mouse models of breast cancer, in which the activation of PPARγ led to rapamycin resistance, whereas rapamycin treatment triggered both the expression and activation of PPARγ [247].

Similar to PPARα, PPARγ regulates autophagy, the main cellular process targeted by mTOR. The activation of PPARγ stimulates autophagy, and its inhibition decreases it [248,249]. In addition to cell cycle arrest, the induction of autophagy is a major effect of PPARγ in cancer cells, as observed in colon [250], breast [251,252], bladder [253], and adrenocortical [254] cancer cell lines. The application of troglitazone and rosiglitazone in MDA-MB-231 breast cancer cells results in the redistribution of microtubule-associated protein 1A/1B-light chain 3 (LC3) and formation of the autophagosome [252]. Liver cells treated with 4-O-carboxymethyl ascochlorin, an agonist of PPARγ, undergo an autophagic response [249]. PPARγ activation in Caco-2 cells and other cell types affects autophagy via positive regulation of PTEN [255,256,257,258,259]. In contrast, the deletion of PTEN results in an increased expression of PPARγ in the liver [260]. In a model of cigarette smoking extract (CSE)-induced inflammation in a human bronchial epithelial cell line (16HBE), PPARγ plays an anti-inflammatory role in an autophagy-dependent manner. In this model, PPARγ activation promotes autophagy by phosphorylating and activating AMPK (see the AMPK and PPARγ section) [261]. Accordingly, the PPARγ agonists TZDs inhibit IGF-1-induced p70S6K phosphorylation at sites specifically phosphorylated by mTOR as well as dephosphorylating a downstream factor, S6K, through AMPK activation [262]. Similarly, PPARγ induces the expression of autophagy-related proteins but reduces inflammatory molecules in aged human dental pulp cells, suggesting a mechanism by which it might maintain pulp homeostasis [263].

Conversely, PPARγ exerts anti-autophagic effects in neurons [264,265] and macrophages [266]. Rosiglitazone downregulates autophagy-related protein expression and improves locomotor function after spinal cord injury (SCI), whereas GW9662, a PPARγ inhibitor, abolishes the protective effect in SCI [265]. The PPARγ agonist 15-PGJ2 counteracts the increased expression of the autophagy-related protein in the ischemic cortex following cerebral ischemia–reperfusion (I/R) injury. Moreover, cerebral I/R injury increases levels of LC3 and cathepsin-B in neurons, whereas 15-PGJ2 decreases them. Thus, PPARγ exerts neuroprotection by inhibiting neuronal autophagy after cerebral I/R injury [264]. Finally, in human monocyte-derived macrophages, PPARγ induces cathepsin L, which is a cysteine protease that has been found to trigger apoptosis and decrease autophagy [266].

## 4. AMPK

CR decreases energy input, which leads to the activation of a signaling cascade to generate fuel and increase longevity. Decreased glucose intake reduces carbon flow through the glycolytic pathway and slows the conversion of ADP to ATP. As a principal cellular energy sensor, AMPK monitors the AMP:ATP and ADP:ATP ratios. Functionally, AMPK is a serine/threonine kinase comprising one catalytic subunit, α, and two regulatory subunits, β and γ. Each of the subunits occurs as different isoforms (α1, α2, β1, β2, γ1, γ2, γ3) allowing for different versions of AMPK in various tissues [267,268]. From nematodes to humans, the kinase activity of AMPK is rapidly increased by the binding of AMP or ADP to the AMPKγ subunit [269]. This binding promotes allosteric activation and the phosphorylation of AMPK by the upstream AMPK kinase and thus also inhibits its dephosphorylation [270]. An alternative activating pathway triggers AMPK in response to increases in cellular Ca^2+^ and involves the Ca^2+^/calmodulin-dependent protein kinase kinase β (CaMKKβ) [271]. Once activated, AMPK promotes ATP preservation by repressing energy-consuming biosynthetic pathways while enhancing the expression or activity of proteins involved in catabolism. This process results in the mobilization of deposited energy to restore the ATP supply [272]. Several downstream factors including CREB-regulated transcriptional coactivator-2 (CRTC2) [273], TBC1D1/AS160 [274,275], PGC-1α [276], and histone deacetylase (HDAC) 5 [277] mediate the impact of AMPK on metabolism. Functionally, AMPK phosphorylates acetyl-CoA carboxylase 1 (ACC1) and ACC2 [278,279], SREBP1c [280], glycerol phosphate acyl-transferase, [281], and HMG-CoA reductase [282], resulting in the inhibition of FA, cholesterol, and TG synthesis while activating FA uptake and β-oxidation. Additionally, AMPK prevents protein biosynthesis by inhibiting mTOR and TIF-IA/RRN3, which is a transcription factor for RNA polymerase I that is responsible for ribosomal RNA synthesis [283]. AMPK also influences glucose metabolism by stimulating both nutrient-induced insulin secretion from pancreatic β-cells [284] and glucose uptake by phosphorylating Rab-GTPase-activating protein TBC1D1, which ultimately induces the fusion of glucose transporter (GLUT)4 vesicles with the plasma membrane in skeletal muscle [285]. AMPK stimulates glycolysis by the phosphorylation of 6-phosphofructo-2-kinase (fructose-2,6-bisphosphatase 2) [286], and in parallel, it inhibits glycogen synthesis through the phosphorylation of glycogen synthase [287]. In the liver, AMPK inhibits gluconeogenesis by inhibiting transcription factors including hepatocyte nuclear factor 4 and CRTC2 [288,289,290]. AMPK also affects the energy balance by regulating circadian metabolic activities and promoting feeding through its action in the hypothalamus [291,292]. It promotes mitochondrial biogenesis via PGC-1α [276] (see the section on mitochondria) and activates antioxidant defenses. AMPK plays a major role in metabolism but is also involved in inflammation, cell growth, autophagy, and apoptosis [293]. Therefore, reducing AMPK signaling exerts a cytostatic and tumor-suppressing effect [294,295].

In *C. elegans*, the lifespan extension effect of CR depends on AMPK [296,297]. Similarly, in *Drosophila*, pathways mediating increased lifespan include AMPK activation [298]. In addition, tissue-specific overexpression of AMPK in muscle and body fat extends the lifespan in *Drosophila*, whereas AMPK RNA interference shortens the lifespan [299]. The link between AMPK and PPARs and their interaction in metabolism regulation in response to CR have been well documented and are discussed below.

### 4.1. AMPK and PPARα

AMPK and PPARα both act as sensors of intracellular energy status and adjust metabolism in response to changes. As noted, AMPK responds to intracellular ATP depletion, whereas PPARα induces the expression of genes encoding enzymes and proteins involved in increasing cellular ATP yields. Moreover, AMPK and PPARα serve as critical regulators of short-term and long-term FA oxidation, respectively, and their activity thus needs to be coordinated. Accordingly, during prolonged fasting, when glucose levels drop and FA levels rise, high intracellular AMP concentrations induce AMPK, resulting in increased mitochondrial FA uptake for β-oxidation. In parallel, the activation of PPARα elevates the maximal FA-oxidizing capacity in the liver [35,37,300,301].

Similar to AMPK, phosphorylation affects the activity of PPARα. Several kinases, including p38, ERK, protein kinase A, and PKC, and AMPK itself can phosphorylate PPARα, which modifies (mainly increasing) its transcriptional activity [302]. However, the activation of p38, which AMPK may execute [303,304], induces the activation of PPARα in some cells while reducing it in others. Additionally, the phosphorylation of PPARα by glycogen synthase kinase, also regulated by AMPK [305], leads to the degradation of PPARα [302,306].

The activation of PPARα by AMPK has been shown in multiple experimental models. In myocytes, either 5-aminoimidazole-4-carboxamide ribonucleotide (AICAR), a synthetic activator of AMPK, or adiponectin, an insulin-sensitizing adipokine, increase FA oxidation gene expression via AMPK-dependent PPARα activation [307,308]. Therefore, the reduced serum levels of adiponectin in people with obesity and T2D may contribute to the observed impairment in PPARα activity [309]. Of note, in muscles, PPARα does not directly interact with AMPK [310]. Similarly, in the left atrial appendage of mixed-breed dogs, the AMPK/PPARα/VLCAD (very long-chain acyl-CoA dehydrogenase) pathway mediates the metformin-triggered reduction of lipid accumulation and increases the β-oxidation of FA [311]. In pancreatic β-cells, glucose represses PPARα gene expression via AMPK inactivation [312,313]. The mechanism of the direct interaction between AMPK and PPARα has been uncovered in hepatocytes. In this pathway, activated AMPKα subunits bind to and activate PPARα, which occurs independently of AMPK activity and is not associated with increased AMP concentration. Instead, the interaction is stimulated by increased MgATP levels. Surprisingly, treatment with AICAR decreases PPARα activity in rat hepatocytes, which is associated with translocation of the AMPKα2 isoform out of the nucleus and is independent of the kinase activity of AMPKα [314]. The contradictory information concerning the interaction between PPARα and the ligands of AMPK likely reflects tissue- and context-specific situations.

One publication has reported that AMPK inhibits PPARα and PPARγ activity [315]. In that study, the AMPK activators, AICAR, and metformin decreased basal and WY-14,643-stimulated PPARα activity in hepatoma cells. Compound C, which is an AMPK inhibitor, increased agonist-stimulated reporter activity and partially reversed the effect of the AMPK activators. The expression of either a constitutively active or dominant-negative AMPKα subunit inhibits basal and WY-14,643-stimulated PPARα activity. The authors postulated that the AMPK inhibition of PPARα and PPARγ may allow for short-term processes to increase energy generation before the cells devote resources to increasing their capacity for FA oxidation [315]. This contradictory report may indicate further that AMPK–PPAR regulation is cell-type specific.

An agonist of PPARα also can activate AMPK, suggesting that the activity regulation between AMPK and PPARα may be reciprocal. On the one hand, fenofibrate induces the phosphorylation and activation of AMPK via the induction of the small heterodimer partner (SHP; an orphan nuclear receptor) and its target genes [316]. On the other hand, WY-14,643 treatment increases the expression of AMPKα1 and α2 mRNA, leading to an increase in AMPK*α* subunit phosphorylation and its enzymatic activity [317]. Moreover, pterostilbene, a bioactive component of blueberries and grapes and an agonist of PPARα, activates AMPK, similarly to AICAR and metformin, and modulates several AMPK-dependent metabolic functions in the rat hepatoma cell line H4IIE [318]. The AMPK-mediated activation of PPARα reverses progressive fibrosis in steatohepatitis [316] by endothelial nitric oxide (NO) synthase (eNOS) phosphorylation in endothelial cells, which suppresses microvascular inflammation and apoptosis [319,320].

### 4.2. AMPK and PPARβ/δ

AMPK and PPARβ/δ, but not PPARα, interact directly and physically in muscle, leading to increased glucose oxidation via the upregulation of lactate dehydrogenase B, which is associated with improved exercise performance [310]. AICAR treatment increases endurance, and the combination of AICAR and GW0742 further potentiates it. The combination significantly increases all running parameters, which is a change that is accompanied by a significant shift to fat as the main energy source with a decline in carbohydrate use during the period near exhaustion [321]. Therefore, agonists of both AMPK and PPARβ/δ are recognized as exercise mimetics [322]. In line with these observations, the deletion of PPARβ/δ specifically in myocytes results in a reduced capacity to sustain running exercise [78].

### 4.3. AMPK and PPARγ

The activation of AMPK by PPARγ agonists has been documented in many cell lines [261,323,324,325,326], in various tissues ex vivo [327,328], and in nonhuman animals [329,330,331] and people [332]. In general, agonists of PPARγ act via AMPK to improve glucose and fat management. Troglitazone causes rapid increases in phosphorylated AMPK and acetyl-CoA carboxylase (ACC) within minutes after injection in rat skeletal muscle, liver, or adipose tissue. Consistently, the drug results in a two-fold increase in 2-deoxy-d-glucose uptake in skeletal muscle through AMPK activation [328]. In addition, rosiglitazone remarkably enhances AMPK-mediated glucose uptake and glycogen synthesis in muscle and adipose tissues [331]. In cardiac muscle, the impact of troglitazone on glucose uptake is triggered via AMPK and eNOS signaling [333]. Rosiglitazone increases the expression and circulating levels of adiponectin and enhances the expression of hepatic adiponectin receptors in mice, which correlates with the activation of the hepatic Sirt1/AMPK signaling system. This signaling enables rosiglitazone to attenuate alcoholic liver steatosis and nonalcoholic steatohepatitis [329,334]. Another TZD, pioglitazone, increases AMPK phosphorylation two-fold and decreases ACC activity and the concentration of malonyl-CoA by 50% in Wistar rat liver. Moreover, pre-treatment with pioglitazone prevents a 50% decrease in AMPK and ACC phosphorylation in the liver and adipose tissue, which can be triggered by a euglycemic–hyperinsulinemic clamp [330]. In endothelial cells, rosiglitazone reduces glucose-induced oxidative stress mediated by NAD(P)H oxidase hyperactivity induced by high glucose via AMPK activation. It also uses AMPK to stimulate eNOS activity to increase NO synthesis [324,325].

Several TZDs have been shown to reduce insulin resistance via AMPK activation [323,327,335,336]. AMPK-mediated pioglitazone signaling results in an increase in insulin-stimulated glucose disposal, enhanced expression of the genes encoding adiponectin receptors, and coding for factors connected with mitochondrial function and FA oxidation in the muscles of patients with diabetes [332]. Rosiglitazone promotes AMPK-mediated insulin secretion via the phosphorylation of the Kir6.2 subunit of the potassium ATP channel in β-cells [336]. The treatment of pancreatic β-cells with TZDs triggers the phosphorylation of AMPK and ACC and increases glucose-stimulated insulin secretion as well as the response of insulin secretion to the combined stimuli of glucose and palmitate [327]. This treatment also affects β-cell metabolism by reducing glucose oxidation, energy metabolism, and glycerolipid/FA cycling [323]. Thus, the role of TZDs in lowering serum insulin levels and in the protection of β-cells is mainly through AMPK [327].

In addition to mediating PPARγ metabolic functions, AMPK mediates the receptor’s anti-inflammatory activities. In bronchial epithelial cells, PPARγ plays a protective role in CSE-induced inflammation, as noted above (see the section on mTOR and PPARγ). CSE administration inactivates AMPK signaling, which is restored by PPARγ agonists. Consequently, the effects of PPARγ agonists on inflammation and also on autophagy can be abolished by AMPK inhibition [261], showing that AMPK is downstream of PPAR in this pathway. AMPK also mediates the anti-inflammatory effect of PPARγ in endothelial cells, in which the LPS-triggered downregulation of toll-like receptor 4 (TLR4) protein expression is inhibited by pioglitazone. LPS also reduces PPARγ expression, which can be partially restored by the knockdown of TLR4. Therefore, TLR4 and PPARγ affect each other via a negative feedback loop, and this interaction depends on the AMPK signaling pathway [326].

As discussed, agonists of PPAR exert physiological effects by modulating the activity of AMPK, which is an important cellular energy sensor. However, their action seems to be, at least in some instances, independent of the activation of the PPARs. In other words, these agonists can activate AMPK by phosphorylation independently of PPAR*γ* or PPARα [316,317,325,328,337]. This idea is supported by a novel TZD, BLX-1002, with no PPAR affinity, which activates AMPK in β-cells and raises cytoplasmic Ca^2+^, thereby enhancing glucose-induced insulin secretion at a high glucose level [335]. Similarly, some agonists of PPAR likely exert some effects independently of PPAR, which is in cooperation with other cellular partners.

AMPK also has been reported to feed back to PPAR*γ*. The expression of either a constitutively active or dominant-negative AMPKα inhibits basal and rosiglitazone-stimulated PPARγ activity. AICAR and metformin inhibit PPRE reporter activity, whereas AMPK inhibitor compound C increases basal and rosiglitazone-stimulated PPARγ activity [315]. 

In brief, there is a very tight interaction between AMPK and PPARs (Figure 4), which involves the factors participating in the metabolic, apoptotic, and anti-inflammatory response to CR.

## 5. Insulin Signaling

Increased glucose levels in serum after food intake promote insulin secretion from pancreatic β-cells, which in turn activates insulin receptors on the surface of target cells. The tyrosine kinase activity of the insulin receptor triggers a signaling cascade starting with the activation of insulin receptor substrates (IRS 1–4) followed by the phosphorylation of PI3K, which is responsible for metabolic actions including PDK1 and Akt activation. Akt occurs in three isoforms (1–3) with Akt2 being essential for glucose homeostasis, whereas Akt1 is important for growth and Akt3 is important for brain development [338]. The Akt-driven inhibition of AS160 phosphorylation induces GLUT4 to translocate to the cell membrane, which promotes glucose transport into the intracellular compartment. Akt also phosphorylates and deactivates glycogen synthase (GS) kinase 3 (GSK3), which stimulates GS and glycogen production. In parallel, it disrupts the CBP/Torc2/CREB complex and consequently inhibits gluconeogenesis. Moreover, Akt activates mTOR, which facilitates protein synthesis, whereas mTORC2 is a critical regulator of Akt [339]. Another Akt regulator, tumor suppressor PTEN, previously mentioned in the context of mTOR, prevents Akt activation and reduces mTOR activity. In line with the above, the inhibition of IGF-1/PI3K/Akt signaling participates in the anti-cancer and DNA-repair activity of CR [340,341,342]. Further, Akt activation leads to the inhibitory phosphorylation of FOXO1, resulting in its nuclear exclusion [343]. Therefore, Akt functions at the crossroads of several pathways responding to CR.

Among other pathways affected by insulin signaling, the most important include mitogen-activated protein kinase (MAPK), which regulates growth; SREBP-1, which promotes lipid and cholesterol synthesis; and the family of FoxO transcriptional regulators, which regulate metabolism and autophagy. In general, insulin signals an abundance of fuels and thus promotes storage and prevents the further production of energy molecules [344,345,346,347].

The beneficial effects of CR have been associated with changes in metabolism, modification of the activity of the insulin/IGF-1 pathway, reduction in fat mass, and increased stress resistance because of FoxO activation [348,349,350]. Insulin release and insulin action seem to play a major role in the control of aging. The modulation of longevity by insulin signaling is supported by the extended lifespan associated with mutations in the insulin/IRS/growth hormone (GH)/IGF-1/FOXO signaling pathways in humans, mice, *C. elegans*, and *Drosophila* [351,352,353,354,355,356,357]. Female, but not male, Igf1r+/− mice live on average 33% longer than their wild-type counterparts [355], and the fat-specific deletion of Igf1r results in an 18% increased longevity in both sexes [351]. Accordingly, GH receptor/binding protein knockout (GHR/BP-KO) mice are characterized by a markedly extended lifespan and show severely reduced plasma IGF-1 and insulin levels, as well as low glucose levels [358,359]. Transgenic Klotho mice, which also have an increased lifespan, are insulin resistant. These findings collectively suggest that aging can be delayed by reducing insulin signaling [360]. It has even been hypothesized that insulin resistance is a physiological protective mechanism against aging and age-related disorders [361].

### 5.1. Insulin Signaling and PPARα

The immense impact of PPARα on glucose homeostasis and insulin signaling is particularly well illustrated by pancreas malfunction and diabetes models. PPARα directly protects pancreatic islets and their function and improves the adaptive response of the pancreas to pathological conditions. PPARα activation during the fed-to-fasted transition affects the regulation of glucose-stimulated insulin release because of the critical role of FA in insulin secretion [362]. In this condition, the activation of PPARα in β-cells increases pancreatic FA oxidation and potentiates glucose-induced insulin secretion [363,364]. In contrast, PPARα activation can oppose insulin hypersecretion elicited by high-fat feeding [365], suggesting that this activation protects pancreatic islets from lipotoxicity. Similarly, in primary human pancreatic islets, PPARα agonist treatment prevents the FA-induced impairment of glucose-stimulated insulin secretion, apoptosis, and TG accumulation, indicating that PPARα mediates the adaptation of pancreatic β-cells to pathological conditions [366]. PPARα participates in a pathway mediating the effect of metformin on glucagon-like peptide-1 (GLP-1) receptor expression in pancreatic islets and on plasma levels of GLP-1 [367], improving glucose management. In addition, PPARα regulates hepatic glucose metabolism by upregulating glycerol-3-phosphate dehydrogenase, glycerol kinase, glycerol transport proteins [368], and pyruvate dehydrogenase kinase 4 during fasting [369], which leads to the promotion of gluconeogenesis over FA synthesis.

In in vivo models of insulin resistance and diabetes, PPARα activation reverses the pregnancy-related augmentation of glucose-stimulated insulin hypersecretion by increasing insulin sensitivity [370]. Similarly, in nondiabetic patients with hypertriglyceridemia and patients with latent diabetes, the improvement in glucose metabolism observed during short-term clofibrate administration may also result from increased insulin sensitivity. Fasting plasma glucose, oral glucose tolerance test results, and immunoreactive insulin in these patients are significantly decreased, which is accompanied by enhanced glucose use and decreased serum TGs and cholesterol [371]. Furthermore, clofibrate in patients with non-insulin-dependent diabetes decreases fasting plasma glucose and insulin levels, and insulin binding to erythrocytes is enhanced because of increased insulin receptor affinity without a change in receptor number [372]. Another study showed that clofibrate ameliorates glucose tolerance in this patient population without changing the number of insulin receptors and that this increased insulin sensitivity occurs via an unknown post-receptor mechanism [373].

Strikingly, chronic fenofibrate treatment completely prevents the spontaneous sequential hypertrophy and atrophy of pancreatic islets from obese diabetes-prone Otsuka Long Evans Tokushima Fatty (OLETF) rats, decreases body weight and visceral fat, and improves insulin action in skeletal muscle [374]. Along the same line of observations, fenofibrate treatment significantly reduces hyperinsulinemia and hyperglycemia in C57BL/6 mice with insulin resistance triggered by a high-fat diet and in a model of genetic insulin resistance (obese Zucker rats) [375]. Similarly, the treatment of db/db diabetic mice with PPARα agonists significantly reduces plasma insulin and insulin resistance, improves hyperglycemia, albuminuria, and kidney glomerular lesions, and causes a 50% reduction in FA oxidation, with a concomitant increase in glycolysis and glucose oxidation [376,377]. PPARα-deficient ob/ob mice with obesity-related insulin resistance develop pancreatic β-cell dysfunction characterized by reduced mean islet surface area and decreased insulin secretion in response to high glucose [366]. Similarly, PPARα KO mice develop marked age-dependent hyperglycemia [366], and after 24-h fasting, severe hypoglycemia accompanied by elevated plasma insulin concentrations [54,378]. However, PPARα KO mice are protected from high-fat diet-induced insulin resistance, which is most likely because of the development of increased adiposity [379]. Of note, PPARα gene variation in humans can affect the age of onset and progression of T2D in patients with impaired glucose tolerance [51,52].

In the liver, the insulin-stimulated activation of Akt induces the phosphorylation of NCoR1 on serine 1460, which selectively favors its interaction with PPARα. Phosphorylated NCoR1 inhibits the activity of PPARα, attenuating oxidative metabolism, whereas it derepresses liver X receptor α (LXRα), resulting in increased lipogenesis [380]. Glucose levels also affect PPARα activity. The exposure of islets or INS(832/13) β-cells for several days to supraphysiological glucose concentrations, which are detrimental to insulin secretion, leads to a 60–80% reduction in PPARα mRNA expression, DNA-binding activity, and target gene expression, which results in diminished FA oxidation and increased TG accumulation that are potentially associated with pancreatic lipotoxicity [381]. Moreover, insulin-activated MAPK and glucose-activated PKC stimulate PPARα transcriptional activity in HepG2 cells [382]. Strikingly, glucose itself can modulate PPARα activity because PPARα binds glucose and glucose metabolites with high affinity, prompting changes in its secondary structure [383]. Overall, based on the effects of PPARα on glucose homeostasis and its important regulatory role in the transition from feeding to fasting, PPARα might be involved in protecting against hypoglycemia during CR.

### 5.2. Insulin Signaling and PPARβ/δ

PPARβ/δ cross-reacts with insulin signaling at several points. At first, PPARβ/δ senses elevated glucose levels. Glucose overload leads to cPLA_2_ activation and the subsequent hydrolysis of arachidonic and linoleic acid and their peroxidation, producing endogenous ligands of PPARβ/δ [384]. In the mouse pancreas, PPARβ/δ represses insulin secretion and the β-cell mass [385]. In adipocytes, it prevents IL-6–dependent STAT3 activation by repressing ERK1/2 and STAT3–Hsp90 association. This effect is thought to prevent cytokine-induced insulin resistance in these cells [386]. Similarly, PPARβ/δ represses IL-6-induced STAT3 activation and suppressor of cytokine signaling-3 (SOCS-3) upregulation in human liver cells and thereby halts the development of insulin resistance [387]. In skeletal muscle cells, PPARβ/δ attenuates ER stress-associated inflammation and prevents insulin resistance in an AMPK-dependent manner [387,388]. Moreover, PPARβ/δ ameliorates hyperglycemia by increasing glucose flux through the pentose phosphate pathway, which enhances FA synthesis. Coupling PPARβ/δ-dependent increased hepatic carbohydrate catabolism and the promotion of β-oxidation in muscle allows PPARβ/δ to regulate metabolic homeostasis and enhance insulin action by complementary effects in distinct tissues [389]. In a primate model of metabolic syndrome, GW501516, an agonist of PPARβ/δ, dose-dependently lowers plasma insulin levels without side effects on glycemic control [390]. GW501516 treatment also markedly improves diabetes by decreasing blood glucose and insulin levels in ob/ob mice [391]. In addition, the treatment of healthy people who are moderately overweight with GW501516 results in a significant reduction in fasting plasma insulin [392], and the dual PPARα/δ agonist GFT505 (elafibranor) improves hepatic and peripheral insulin sensitivity in men with abdominal obesity [393].

### 5.3. Insulin Signaling and PPARγ

PPARγ is an established regulator of insulin sensitivity, making it an excellent drug target (Figure 5). TZDs form a class of PPARγ agonists that reverse insulin resistance in liver and peripheral tissues, reducing plasma glucose through specific PPARγ activation. Troglitazone was the first TZD approved for this use, but it was withdrawn from the market following reports of serious hepatotoxicity in some patients. TZDs not only improve insulin sensitivity but also preserve pancreatic β-cell function, thus reducing T2D incidence, as demonstrated in clinical trials of T2D prevention in high-risk people [394,395].

PPARγ exerts its insulin-sensitizing properties in several ways. First, it generates functional WAT, which is required for proper glucose homeostasis because lipodystrophy is associated with severe insulin resistance [396]. An early consequence of PPARγ activation that precedes decreased blood TG and glucose is the stimulation of TG production and a reduction in circulating free FA because of FA retention in fat rather than muscle and pancreas. Consequently, increased fat mass triggered by PPARγ activation results in improved glycemic control [397]. Accordingly, the level of insulin sensitization following PPARγ activation is correlated with the reduction in lipid accumulation in skeletal muscle [398]. Furthermore, in mice fed a high-cholesterol/fructose diet, the selective PPARγ agonist pioglitazone improves insulin sensitivity by affecting its signaling pathway, as measured by induction of IRS-2 expression and increased phosphorylation of Akt and GSK-3β [399]. In fact, PPARγ induces the expression of several proteins in the insulin-signaling pathway, including IRS-1 [400], IRS-2 [401], the p85 subunit of PI3K [402], and Cbl-associated protein (CAP) [403,404]. In 3T3-L1 adipocytes and diabetic rodents, PPARγ directly binds the promoter of the *Cap* gene. Increased CAP expression results in increased insulin-stimulated c-Cbl phosphorylation [403] and consequently in increased glucose uptake [405]. The activation of PPARγ in muscle cells and adipocytes increases the expression and translocation of GLUT1, GLUT2, and GLUT4 to the cell membrane, thus increasing glucose uptake and consequently reducing glucose plasma levels [406,407,408]. In parallel, PPARγ regulates the expression of genes responsible for glucose disposal [400,401,402,403,404].

An important contributor to the insulin-sensitizing effect of PPARγ ligands is the suppression of local and systemic cytokine production. TZD as treatment for patients with obesity and without diabetes reduces circulating levels of inflammatory cytokines and other pro-inflammatory markers, which are accompanied by improved insulin sensitivity [409]. Moreover, hepatic PPARγ reduces the expression of SOCS-3, which has been suggested to play a crucial role in linking inflammation and hepatic insulin resistance [399]. SOCS-3 promotes the ubiquitination and degradation of IRS-2 and thus modulates insulin signaling [410,411]. In vitro studies have confirmed that PPARγ agonists may also exert their antidiabetic activities by counteracting the negative effects of TNFα [412]. In addition, PPARγ elevates blood levels of adipocytokines, such as adiponectin, which are present at low concentrations in the plasma of patients with T2D. The increased adiponectin levels improve insulin sensitivity and free FA oxidation and reduce glucose production in the liver [413,414]. The signaling of PPARγ involves the previously mentioned executor of insulin signaling, FOXO. FOXO1 acts as a transcriptional repressor of *Pparγ* by binding to its promoter and may reduce PPARγ transcriptional activity through a transrepression mechanism involving direct protein–protein interaction between FOXO1 and PPARγ. This interaction seems to be a crucial part of the pathway responsible for insulin sensitivity in adipocytes [415,416,417]. Moreover, insulin signaling in the liver directly affects PPARγ, as Akt2 stimulates the expression and activity of PPARγ in hepatocytes, resulting in elevated aerobic glycolysis and lipogenesis [260].

As a result of this effect on regulatory pathways, TZDs improve insulin sensitivity, glucose tolerance, and the lipidemic profile in T2D as well as in obesity without diabetes [418]. Dominant-negative mutations in human PPARγ can lead to severe metabolic syndrome, insulin resistance, and diabetes at an unusually young age [419,420], and several point mutations in the PPARγ gene are associated with severe insulin resistance (with or without T2D) and familial partial lipodystrophy phenotypes [421,422,423,424,425]. Both partial and generalized lipodystrophies have consistently been associated with insulin resistance in animals and humans [426]. Therefore, it is likely that the dramatic reduction in limb and gluteal fat found in subjects with PPARγ mutations contributes to their insulin resistance. In addition, the residual adipose tissue in these individuals is dysfunctional, likely resulting in unregulated FA fluxes and impairing insulin action in skeletal muscle and liver [420]. Of interest, lipodystrophic, WAT-specific PPARγ KO mice show an increased expression of PPARγ in the liver, which promotes insulin sensitivity [427,428]. In this context, it is important to note that insulin sensitivity declines with age in humans and is accompanied by a lower expression of PPARγ in preadipocytes [429]. Hence, FA metabolism becomes altered with aging in preadipocytes, which correlates with increased susceptibility to lipotoxicity and impaired FA-induced adipogenesis. In line with these observations, PPARγ, PPARα, and RXR levels are all increased in the liver of GHR-KO long-lived animals [131]. Thus, the enhanced insulin sensitivity in GHR-KO mice may be the result of the increased hepatic activity of PPAR family members.

In addition to TDZs, several other PPARγ agonists influence insulin and glucose management. FMOC-L-Leucine (F-L-Leu) is a partial agonist that selectively activates some PPARγ pathways. F-L-Leu improves insulin sensitivity in normal, diet-induced glucose-intolerant mice and in diabetic *db*/*db* mice, yet it has a lower adipogenic activity [430]. Of interest, INT131 besylate, which is a potent non-TZD-selective PPARγ modulator, induces a dose-dependent reduction in fasting plasma glucose without evoking fluid retention or weight gain, which are both unwanted side effects often triggered by TZDs [431]. In addition, food-derived active compounds may contribute to the management of glucose levels. The plant polyphenols quercetin and kaempferol serve as weak partial agonists of PPARγ and increase insulin sensitivity and glucose uptake via PPARγ agonism [432,433]. Another compound, 13-oxo-9(*Z*),11(*E*),15(*Z*)-octadecatrienoic acid (13-oxo-OTA), a linolenic acid derivative in the extracts of tomato (*Solanum lycopersicum*), Mandarin orange (*Citrus reticulata*), and bitter gourd (*Momordica charantia*), modulates gene expression and the production of adiponectin through PPARγ in adipocytes [434]. The reduction of PPARγ activity by antagonists improves the metabolic profile in mice [435,436], and haplodeficient *Pparγ^+/−^* mice exhibit increased insulin sensitivity compared with their wild-type littermates [437,438]. These animals are characterized by reduced fat deposits and lower levels of TG accumulation and lipogenesis in WAT, skeletal muscle, and liver [439]. Similarly, genetic variants Pro(12)Ala (heterozygotes) and Ala(12)Ala (homozygotes) of PPARγ, which result in decreased receptor activity, are associated with leanness and improved insulin sensitivity [440,441,442]. A complex U-shaped curve has been proposed to characterize the relationship between PPARγ activity and insulin sensitivity [99]. 

Altogether, overwhelming evidence points to an important role for all three PPARs in insulin signaling and glucose level management, and to several compounds with similar potential, including some that block the endogenous ligand-induced activation of PPAR*γ* for the treatment of the metabolic syndrome and T2D [436,443,444].

## 6. Sirtuins

As already mentioned, a CR-related decrease in energy levels leads to the activation of several signaling cascades. Decreased glucose intake reduces the flow of carbon through the glycolytic pathway and the regeneration of ATP from ADP, which eventually alters the NAD+:NADH ratio. This shift activates SIRTs, which serve as both energy sensors and transcriptional effectors by acting as NAD+-dependent HDACs. In addition to CR and fasting, exercise activates SIRTs [445,446], which are remarkably conserved and can even be found in archaebacteria [447]. Originally categorized as class III HDACs, SIRTs are involved in the proper functioning of nucleic acids including DNA repair, homologous recombination, and DNA deacetylation, and they promote transcriptional gene silencing [448,449].

The seven subtypes of SIRTs (SIRT1–7) in mice and humans vary in their cellular distribution and function. SIRT1–SIRT3, SIRT5, and SIRT6 catalyze deacetylation, whereas SIRT4 and SIRT6 have ADP-ribosylation capacity. In addition to histones, SIRT substrates include several transcriptional regulators, such as the nuclear factor kappa-light-chain enhancer of activated B cells (NF-κB), p53, FOXO, and PGC-1α, but also enzymes, including acetyl coenzyme A synthetase 2 (AceCS2), long-chain acyl-coenzyme A dehydrogenase (LCAD), HMGCS2, superoxide dismutase 2, and structural proteins, such as α-tubulin [450,451,452,453,454]. Therefore, SIRTs influence a wide range of cellular processes including circadian clocks, cell cycle, mitochondrial biogenesis, and energy homeostasis, and on the whole-body level regulate aging, apoptosis, inflammation, and stress resistance [455,456].

SIRT1 is the most thoroughly investigated mammalian SIRT and is closely involved in metabolism. Studies in *S. cerevisiae* have shown that an extra copy of the Sir2 gene, a yeast homolog of mammalian Sirt1, increases lifespan in a dose-dependent manner [457,458], and the deletion of this gene shortens lifespan [457]. In yeast and *Drosophila*, a lack of Sir2 and dSir2, respectively, prevents CR-associated life extension [459,460,461]. SIR2, a yeast analog of Sirt1, assists in DNA repair and regulates genes that change expression with age [462].

The most important metabolic regulator affected by SIRT1 is PGC-1α, which is activated by SIRT1-mediated deacetylation [463,464]. Deacetylated PGC-1α increases hepatic gluconeogenic activity [463], whereas in muscle and BAT, PGC-1α enhances mitochondrial activity. The activity of PGC-1α translates into increased exercise capacity and thermogenesis, leading to protection against the onset of obesity and associated metabolic dysfunction [465]. The deacetylation of PGC-1α by SIRT1 depends on cellular NAD^+^ levels, so the status of cellular energy affects PGC-1α activity, which adapts cellular energy production through mitochondrial biogenesis and function. Furthermore, among the SIRT1 substrates are factors that control cell proliferation and apoptosis, including the tumor suppressor protein p53 [466]. The overexpression of SIRT1 hinders p53 transcriptional activity and p53-dependent apoptosis triggered by DNA damage and oxidative stress, whereas the overexpression of dominant-negative SIRT1 can enhance cellular stress responses [466,467].

SIRTs also regulate the activity of the FOXO family of transcription factors [468,469], which affects cell differentiation, transformation, and metabolism as well as plays an important role in CR and longevity regulation [470,471,472]. SIRT1-mediated deacetylation of FOXO1 affects its shuttling between the nucleus and cytoplasm, influencing the expression of FOXO1 target genes and promoting gluconeogenesis and glucose release from hepatocytes [473]. The deacetylation of FOXO3a by SIRT1 increases its translocation from the cytoplasm to the nucleus [474] and its DNA-binding activity. In the nucleus, SIRT1 and FOXO3a form a complex that induces cell-cycle arrest and resistance to oxidative stress, also inhibiting the ability of FOXO3a to induce apoptosis [473]. SIRT1 directly suppresses the expression of uncoupling protein 2 (UCP2), leading to an improved coupling of mitochondrial respiration and ATP synthesis, which induces insulin secretion in β-cells [475]. Confirming the role of SIRT1 in the pancreas, SIRT1^−/−^ mice are characterized by impaired insulin secretion in response to glucose compared with wild-type littermates [475,476]. Conversely, β-cell-specific SIRT1-overexpressing mice exhibit improved glucose tolerance and an enhanced glucose-stimulated insulin secretion [476]. In contrast, SIRT4 has an inhibitory effect on amino acid-stimulated insulin secretion. It represses the activity of glutamate dehydrogenase, reducing α-ketoglutarate production and ATP generation, which are known to activate insulin secretion in pancreatic β-cells [477]. To avoid amino acid-stimulated insulin secretion during CR, when amino acid turnover increases, CR decreases SIRT4 activity, which is opposite to the induction of SIRT1 activity during CR [477]. Considering that NAD^+^ controls the activities of both SIRT4 and SIRT1, their opposing effects on insulin secretion are surprising, and the full implications remain to be understood.

The role of other SIRT family members has been less investigated; thus, their function is less well known. SIRT2 is localized mainly in the cytoplasm, where it deacetylates tubulin filaments, HOXA10, and FOXO [478,479,480,481]. It takes part in multiple processes including cell cycle regulation [482], lifespan extension [457,483], and glucose and lipid metabolism [451,484]. SIRT3 plays an important role in mitochondria maintenance by acting as a deacetylase for a number of mitochondrial matrix proteins [485,486]. During a prolonged fast, SIRT3 activates FA breakdown by the deacetylation of LCAD [453] and stimulates the production of ketone bodies by activating HMGCS2 [452]. Of note, SIRT3 is genetically linked to lifespan in the elderly [487].

SIRT4 has ADP-ribosylation activity and in addition to blocking amino acid-induced insulin secretion [477], it regulates FA oxidation in hepatocytes and myocytes [488]. Both SIRT4 and SIRT5 show mitochondrial localization [477,489]. SIRT6 resides in the nucleus and is involved in genomic DNA stability and promotes the repair of DNA double-strand breaks [490]. SIRT6-deficient mice present a shortened lifespan and a degenerative aging-like phenotype [491]. In contrast, transgenic male mice overexpressing SIRT6 display lower serum levels of IGF-1, higher levels of IGF-1-binding protein, and modified phosphorylation patterns of different components of the IGF-1 signaling pathway, possibly contributing to about a 15% increase in lifespan when compared to wild-type animals [492].

SIRT1 and SIRT6 are both connected with CR-triggered extension of ovarian lifespan, which is mediated by the inhibition of the transition from primordial to developing follicles and by a delay in the growth phase of follicles to preserve the supply of germ cells [493]. SIRT7 is associated with nucleoli and is implicated in the activation of transcription by RNA polymerase I [494] as well as the repair of double-strand breaks by non-homologous end-joining [495]. SIRT7 knockout mice display features of premature aging [495]. SIRT1, SIRT6, and SIRT7 facilitate DNA repair, and this repair slows the aging process. During CR, except for SIRT4, the expression and activity of SIRTs are increased in many tissues, including adipose and brain [496,497,498], heart [499,500], and liver [501]. SIRT1 mediates a broad array of physiological effects of CR. The overexpression of SIRT in worms and flies increases their lifespan [460,461], and accordingly, mutants of SIRT do not show lifespan extension by CR [459,502]. Moreover, transgenic mice overexpressing SIRT1 show phenotypes similar to those of CR mice [503]. The previously mentioned role of yeast Sir2 in lifespan is particularly critical in the context of CR.

Resveratrol, a polyphenolic compound present in, for example, red grapes and wine, stimulates SIRT1 expression, resulting in extended lifespan and health span in treated animals [504]. SIRT1 activation by resveratrol mimics CR and delays aging in a wide range of organisms, from *S. cerevisiae* [505] to *C. elegans* to *Drosophila* [506] and mice [507]. Resveratrol is considered one of the mimetics not only of CR but also of exercise [504,508]. In mice, resveratrol inhibits gene expression profiles associated with muscle aging and age-related cardiac dysfunction [509]. The compound protects mice against diet-induced obesity and the associated insulin resistance through enhanced mitochondrial function mediated by PGC-1α [465].

### 6.1. SIRT and PPARα

During fasting, SIRT4 levels decrease in the liver and SIRT4-null mice display an increased expression of hepatic PPARα target genes associated with FA catabolism [510], indicating that PPARα is a negative downstream target of SIRT4. In contrast, the hepatocyte-specific deletion of SIRT1 impairs PPARα signaling and decreases FA β-oxidation, whereas the overexpression of SIRT1 induces the expression of PPARα targets (Figure 6). In fact, SIRT1 interacts with PPARα and is required to activate PGC-1α by deacetylation. Of note, SIRT1-deacetylated PGC-1α can function as a coactivator in PPARα complexes controlling the expression of several metabolic genes. Therefore, SIRT1 activates PPARα to promote FA oxidation in the liver [511]. Similarly, in the heart, PPARα and SIRT1 modulate FA metabolism [512]. Both PPARα and SIRT1 are upregulated by pressure overload in the heart. The haploinsufficiency of either PPARα or SIRT1 reduces pressure overload-induced cardiac hypertrophy and failure, whereas the simultaneous induction of PPARα and SIRT1 aggravates cardiac dysfunction. PPARα and SIRT1 jointly suppress genes involved in mitochondrial functions that are controlled by the estrogen-related receptors (ERRs). PPARα binds and recruits SIRT1 to the ERR response element. In doing so, it represses ERR target genes in an RXR-independent manner. Suppression of the ERR transcriptional pathway by PPARα/SIRT1 also is a physiological response to fasting [513,514,515].

### 6.2. SIRT and PPARβ/δ

PPARβ/δ markedly increases the transcription [516] and protein levels of SIRT1 [517], whereas PPARα and PPARγ do not stimulate SIRT1 expression [516]. Moreover, PPARα and PPARβ/δ promote osteogenic differentiation in an SIRT1-dependent manner [518,519], and PPARγ prevents it [520]. Based on the PPARβ/δ–SIRT1 interaction, a reasonable inference is that during starvation, increased levels of lipolysis-derived free FAs activate PPARβ/δ. This activation leads to enhanced SIRT1 expression, promoting the deacetylation of factors involved in mitochondrial beta-oxidation and cell survival [516]. The regulation of SIRT1 and PPARβ/δ activity operates bidirectionally. First, in human HaCaT keratinocytes, GW501516 modulates inflammation by acting via AMPK and SIRT1 to reduce TNFα-induced IL-8 mRNA levels and NF-κB DNA-binding activity [517]. Second, the upregulation of SIRT1 by PPARβ/δ attenuates premature senescence in angiotensin (Ang) II-treated human coronary artery endothelial cells. Resveratrol can mimic the action of PPARβ/δ on Ang II-induced premature senescence and reactive oxygen species (ROS) generation [521].

### 6.3. SIRT1 and PPARγ

The interaction between PPARγ and SIRT1 is twofold (Figure 6). PPARγ inhibits SIRT1 expression by binding to the *Sirt1* promoter, and PPARγ also directly interacts with and inhibits SIRT1 activity, forming a negative feedback loop [522]. Pioglitazone prevents NF-κB activation through a reduction in p65 acetylation via the AMPK-SIRT1/p300 pathway [523], whereas SIRT1 represses PPARγ actively via docking with two of its cofactors, NcoR and SMRT [524]. Conversely, the treatment of 3T3-L1 adipocytes with resveratrol represses the expression of PPARγ target genes as well as of PPARγ itself. Furthermore, this treatment increases targeting of the PPARγ protein to the ubiquitin–proteasome system for degradation [525]. Hence, SIRT1 acts as a corepressor of PPARγ-mediated transcription. From a functional point of view, the repression of PPARγ by SIRT1 counters adipogenesis, and the upregulation of SIRT1 triggers lipolysis and the release of fat from differentiated adipocytes [22,524]. Following food withdrawal, SIRT1 promotes fat mobilization by repressing PPAR*γ*, which reduces the expression of genes mediating fat storage [524]. In line with these observations, SIRT1^+/−^ mice show a compromised mobilization of FAs from adipose tissue during fasting [524].

## 7. Major Outcomes of CR

### 7.1. Oxidative Stress Reduction

ROS are generated as a by-product of cellular respiration, contributing to the accumulation of oxidative damage and the formation of a range of oxidation products of different macromolecules including lipids, proteins, and nucleic acids [526]. A small amount of ROS is normally beneficial because it plays an important role in cellular processes such as cell cycle progression, the regulation of signaling pathways in response to intra- and extracellular stimuli, and inflammation [527]. However, high uncontrolled levels of ROS are detrimental.

During oxidative stress, the sustained production of ROS and reactive nitrogen species leads to a perturbed equilibrium between pro-oxidants and antioxidants. Consequently, macromolecules, organelles, and cells are altered, and if much damage accumulates, necrotic or apoptotic cell death occurs. The “free radical theory” of aging [528] proposes that the generation of oxidative stress is a major factor contributing to the onset of the aging process and age-related diseases. Therefore, the mammalian lifespan is reduced in relation to the mitochondrial production of oxidizing free radicals [527]. CR likely exerts its diverse benefits through reducing ROS levels and suppressing age-related oxidative stress while supporting the antioxidant defense system [529,530,531]. CR diminishes the impact of ROS through three processes: reduction of oxygen free-radical generation by slowing metabolism, the acceleration of ROS neutralization, and stimulation of the repair of ROS-damaged molecules [532,533,534,535,536].

The oxidative stress-related role of PPARs is first suggested by their name: they were first identified as receptors stimulating peroxisome proliferation. Peroxisomes have oxidative functions that involve use of molecular oxygen and that yield hydrogen peroxide (H_2_O_2_). The name of these organelles comes from their hydrogen peroxide-generating and scavenging activities. In addition to the conversion of ROS, peroxisomes play a key role in metabolism, catabolizing very long-chain FAs, branched-chain FAs, bile acid intermediates (in the liver), D-amino acids, and polyamines. The induction of oxidative stress is associated with the downregulation of PPARs, which also occurs during aging [140,537,538]. The reduced expression of PPARα in aging [137,539] has been attributed to increased oxidative stress, and CR has been suggested to prevent this decrease through antioxidative action [140].

PPARα-deficient mice present increased oxidative stress at an earlier age than aged-matched wild-type controls [137]. In fact, the administration PPARα agonists to aged mice restores the cellular redox balance, documented by reduced tissue lipid peroxidation, reduced spontaneous inflammatory cytokine production, and the elimination of constitutively active NF-κB [137]. WY-14643 and fenofibrate protect mice from acetaminophen-induced hepatotoxicity by upregulating UCP-2, which is a PPARα target gene that reduces the generation of mitochondrial ROS [540]. In a gentamicin-induced model of ROS production, different types of PPARα and PPARγ agonists (fenofibrate, pioglitazone, tesaglitazar) provide protection from toxicity. These ligands prevent oxidative stress by increasing the expression of genes controlling ROS production and detoxification (SOD1, glutathione peroxidase 1 (GPx1), CAT, UCP-2), which will restore the ratio of reduced to oxidized glutathione and prevent apoptosis [541].

PPARγ directly modulates the expression of several antioxidant and pro-oxidant enzymes as well as oxidative stress-related proteins. It transcriptionally regulates mouse, rat, and human catalase, which is a major antioxidant enzyme converting H_2_O_2 _to O_2 _and H_2_O [542,543]. Similarly, it directly regulates the expression of manganese superoxide dismutase (MnSOD), which performs the dismutation of O_2_^−^ to O_2_ and H_2_O. Conversely, heart-specific PPARγ knockout mice show downregulated levels of MnSOD in cardiac muscle with a consequent increase in O_2_^−^ levels, suggesting that PPARγ protects cardiomyocytes from oxidative damage [544]. In human skeletal muscle cells, the TZD-mediated activation of PPARγ induces GPx3 and protects against oxidative stress [545] because GPx reduces H_2_O_2_ to H_2_O and O_2_ and scavenges for oxidized lipids. PPARγ also represses the expression of inducible NO synthase (iNOS) and stimulates eNOS [546,547,548,549,550]. These enzymes produce NO from arginine, which forms highly reactive peroxynitrite when it reacts with O_2_^−^. In mice with an endothelial-specific knockout of PPARγ, aortic segments release less NO than those from controls, and this reduced expression correlates with an increase in oxidative stress parameters [548].

Cyclooxygenase-2 (COX-2) is an inducible form of cyclooxygenase that contributes to the metabolism of arachidonic acid-forming prostaglandin H2 [551,552], which requires the presence of free radicals and may produce O_2_^−^, contributing to oxidative stress. PPARγ regulates COX-2 expression, but both induction [553,554] and reduction [555,556] in PPARγ expression have been reported, leaving the issue for further investigation. In rats, the activation of PPARγ by oral intake of rosiglitazone upregulates UCP-2 [557], which protects against oxidative stress by preventing O_2_^−^ accumulation in the mitochondria and facilitating the export of mitochondrial ROS to the cytosol [558]. Moreover, a major target gene of PPARγ, CD36, may act as a scavenger receptor that mediates the recognition and internalization of oxidized lipids [559,560,561]. Finally, PPARγ also has been shown to protect cardiomyocytes and glial cells from oxidative stress-induced apoptosis by increasing Bcl-2 [562,563].

In addition to direct transcriptional regulation, PPARγ can modulate the inflammatory and oxidative status by acting on transcription factors such as NF-κB [547,550,564,565]. NF-κB action is usually pro-inflammatory and pro-oxidant, inducing the expression of genes encoding the inflammatory cytokines sIL-1β, IL-6, and TNFα, as well as the pro-inflammatory enzymes COX-2 and iNOS, but it may also regulate the expression of superoxide dismutases and other anti-inflammatory genes [552,566,567,568,569,570]. PPARγ reduces NF-κB activities in various ways: (1) by transrepressing NF-κB activation through forming a repressor complex in the promoter of NF-κB-target genes; (2) by directly binding with NF-κB [547,550,564]; or (3) by catalase-mediated H_2_O_2_ reduction, which activates NF-κB [542,543,571]. Conversely, NF-κB negatively regulates PPARγ transcriptional activity via a mechanism that requires the presence of HDAC3 [572,573].

Of note, PPARγ interacts with a major regulator of the antioxidative response, the nuclear factor erythroid 2-related factor 2 (NRF2). NRF2 is a redox-sensitive transcription regulator that plays a vital role in cryoprotection against oxidative and electrophilic stress as well as in inflammation suppression [574]. NRF2 targets multiple genes, including NADPH-generating enzymes [575], glutathione S-transferases [576], CD36 [560,577], and HO-1 [578,579] and it stimulates the production of defense proteins during oxidative stress. NRF2 also induces PPARγ expression by binding the upstream promoter region of the nuclear receptor [580,581]. Conversely, PPREs have been identified on the NRF2 gene promoter [576,581], confirming a positive feedback loop between PPARγ and NRF2. Therefore, the ability of PPARs to extinguish oxidative stress overlaps with CR effects.

### 7.2. Mitochondrial Function

One of the several theories tightly connected with the effects of ROS is the “mitochondrial theory of aging”, which proposes that mitochondria are the critical component in the aging process. In fact, mitochondrial DNA damage and dysfunction increase with aging and are associated with a vast number of pathologies. Defective mitochondria determine the turnover not only of the organelles themselves but also whole cells, resulting in the acceleration of aging [527,582,583]. Aging has been linked to a reduced capacity for oxidative phosphorylation in the muscle and heart, most likely because of a decline in mitochondrial content and/or function [584,585,586]. Accordingly, young individuals have higher respiratory function compared to the elderly [587,588,589]. Disturbed mitochondrial electron transfer increases the likelihood of electron leakage and ROS production. Consequently, components of the electron transport chain and mitochondrial DNA become damaged, leading to further increases in intracellular ROS levels and a decline in mitochondrial function. Since mitochondrial DNA is spatially close to the source of ROS production, it is thought to be particularly vulnerable to ROS-mediated lesions [528,590].

An interesting feature of CR, one associated with ROS and changes in metabolism, is mitochondria biogenesis, which is relatively high in various tissues such as in the brain, heart, liver, and particularly the BAT of mice [498,591]. It is associated with activation of the master regulator of mitochondrial biogenesis, PGC-1α [428,592,593]. PGC-1α is expressed at a high level in BAT, heart, skeletal muscle, brain, and kidney, whereas its expression is low in the liver and very low in WAT [594]. Various physiological stimuli highly induce PGC-1α in different organs. It is increased in BAT by cold exposure and in skeletal muscle by exercise and decreased ATP level, whereas in the liver, it is mostly affected by CR [595]. When ectopically expressed in fat or muscle cells, PGC-1α strongly increases mitochondrial biogenesis and oxidative metabolism, which correlates with an increase in mitochondrial DNA and the expression of multiple mitochondrial genes [595,596]. To prevent a mitochondrial biogenesis-associated increase in ROS levels, PGC-1α also induces expression of the antioxidant genes *GPx1* and *MnSOD* [597]. One hypothesis regarding the beneficial outcomes of CR proposes is that CR preserves mitochondrial function by maintaining protein and DNA integrity through decreasing mitochondrial oxidant emission and increasing endogenous antioxidant activity [598,599]. Its impact on mitochondria biogenesis remains a matter of discussion [600,601].

In addition to affecting mitochondria biogenesis, PGC-1α also influences metabolism. It mediates a fasting-induced increase in FA metabolism and the downregulation of pyruvate dehydrogenase, which is part of the mitochondrial pyruvate dehydrogenase complex that catalyzes the reaction representing pyruvate entry into the tricarboxylic acid cycle. In PGC-1α knockout mice, pyruvate dehydrogenase fails to adapt to CR, and the ability of the mice to endure prolonged starvation is decreased [602]. PGC-1α knockout mice also show a reduced content of mitochondrial electron transport chain proteins in skeletal muscle [603,604]. The activity of PGC-1α is directly regulated by the energy sensors SIRT1 and AMPK [276,463]. Functionally, the transcriptional activity of PGC-1α relies on its interactions with transcriptional factors for controlling FA metabolism. Of note, all three PPAR isotypes are subject to transcriptional coactivation by PGC-1α and are major executors of PGC-1α-induced regulation [72,594,605,606].

Evidence has accumulated for an important role of PPARs in maintaining healthy mitochondria. Agonists of PPARα and PPARγ modulate mitochondrial fusion and fission in neurons, leading to a better response to oxidative stress and neuron protection [607]. The abnormal expression of PPARα is linked to an altered mitochondrial structure and metabolic function, with an increase in number of cristae, and myocardial damage and fibrosis in PPARα knockout mice [608]. Through its key role in FA β-oxidation, PPARα is inevitably associated with mitochondrial function [35,609]. The activation of PPARα rescues mitochondrial depletion and failure to oxidize FA in the liver-specific class 3 PI3K-deficient mice. In this model, PPARα stimulates mitochondrial biogenesis and lipid oxidation by the inhibition of HDAC3 [610].

In addition, fenofibrate ameliorates insulin resistance accompanied by an improved mitochondrial oxidative capacity in pediatric burn patients [611]. Fenofibrate and gemfibrozil also reduce mitochondrial membrane potential depolarization, resulting in apoptosis inhibition in lymphoblast cells in Batten disease [612]. Pretreatment of rats with gemfibrozil prior to global cerebral I/R results in neuroprotection by modulating mitochondrial biogenesis and apoptosis [613]. WY-14,643 and fenofibrate protect mice from acetaminophen-induced hepatotoxicity by upregulating UCP-2, which is a PPARα target gene that reduces the generation of mitochondrial ROS [540]. However, fibrates may also trigger mitochondrial dysfunction because they inhibit the activity of mitochondrial respiratory chain complex I in rat skeletal muscles [614]. Moreover, gemfibrozil and WY-14,643 alter mitochondrial energy production by promoting mitochondrial permeability transition, as documented by membrane depolarization and calcium-induced swelling, which inhibits the oxidative phosphorylation and ATP synthesis in the rat liver [615]. Finally, chronic treatment with WY-14,643 impairs myocardial contractile function while decreasing mitochondrial respiratory function and increasing mitochondrial uncoupling in rats [616].

PPARβ/δ has been reported to be essential for the exercise-induced increase in the number of muscle mitochondria [617]. In high-fat–fed C57BL/6 mice, the administration of GW501516, a PPARβ/δ agonist, increases the metabolic rate, reduces fatty liver, decreases lipid accumulation, and increases mitochondrial biogenesis in the muscle [391]. In C2C12 muscle cells, GW501516 induces the mRNA expression of UCP-1, UCP-2, and UCP-3 [618], which are responsible for uncoupling mitochondrial respiration [619]. Accordingly, muscle-specific PPARβ/δ KO mice show a reduction in the expression of genes connected with energy uncoupling, mitochondrial electron transport chain, FA uptake, and catabolism. They develop obesity and diabetes with aging and present a mild defect in glucose metabolism when challenged with a high-fat diet [72,78]. The overexpression of either wild-type PPARβ/δ or constitutively active VP16-PPARβ/δ in muscle results in fiber-type switching and augmented capacity for mitochondrial pyruvate oxidation, which is accompanied by an induction of mitochondria numbers [80,310,620]. This phenotype resembles that of the muscle-specific overexpression of PGC-1α [621] and accords with PGC-1α acting as a co-activator of PPARβ/δ to control mitochondrial biogenesis and muscle fiber-type plasticity [72,78,80]. Furthermore, the expression in BAT of a constitutively active PPARβ/δ results in the increased expression of genes involved in FA oxidation and lipolysis and in energy uncoupling in the mitochondria. However, some studies have shown that the impact of PPARβ/δ on energy metabolism in muscles does not necessarily involve *de novo* mitochondrial biogenesis [622,623].

PPARγ and TZDs also have roles in mitochondrial structure and function. Transgenic mice overexpressing PPARγ2 have a significantly increased expression of mitochondrial UCP-1, elevated levels of PGC-1α, and reduced mitochondrial ATP concentrations in their subcutaneous fat [624]. The overexpression of cardiac PPARγ results in the production of mitochondria with a distorted architecture of the inner matrix and disrupted cristae [625]. Rosiglitazone treatment of ob/ob mice leads to mitochondrial remodeling, enhanced oxygen consumption, and increased energy expenditure in WAT [626]. Moreover, in adipose tissues of patients with diabetes, agonists of PPARγ increase the relative amounts of mitochondria and mitochondrial DNA copy number and stimulate the expression of factors involved in mitochondrial biogenesis, respiratory complexes I–IV, and FA oxidation [627,628]. A similar response has been described in differentiated 3T3-L1 and C3H/10T1/2 adipocytes treated with rosiglitazone, which showed increased mitochondrial biogenesis, oxygen consumption, and mitochondrial citrate synthase activity [629]. Finally, rosiglitazone also protects T lymphocytes from apoptosis by preventing the loss of mitochondrial membrane potential [630]. The mitochondria-related impact of TZDs has been identified as a basis for their neuroprotective effect [607,631,632,633,634]. Of interest, TZDs can also exert PPARγ-independent effects on mitochondrial respiration, leading to changes in glycolytic metabolism and fuel substrate specificity [635,636]. Taken together, the evidence strongly suggests that these three PPARs contribute to the maintenance of mitochondria in a tissue-specific manner.

### 7.3. Reduction of Inflammation

The “inflammation hypothesis of aging” posits a molecular mechanism of aging based on inflammation. Inflammation is a complex defense reaction to insult and both physiological and nonphysiological stress, which is induced by agents such as chemicals, drugs, or microbial entities. Inflammation responses are activated by well-coordinated, sequential events controlled by humoral and cellular reactions. Elevated tissue levels of TNFα, IL-1, and IL-6, among other pro-inflammatory mediators, have been observed in experimental animal models of inflammation. With aging, inflammatory responses may be overactive or even cause damage, resulting in pathological conditions [14].

During aging, a shift occurs in the ratio of naive to memory T cells, with associated changes in the cytokine profile in favor of inflammatory cytokines such as TNFα, IL-1, IL-6, INFγ, and transforming growth factor β [637,638,639,640]. There is also a progressively higher dysregulation of immune cells and pro-inflammatory responses. Macrophages from old mice produce more prostaglandin E2 than those from young mice because of higher COX-2 activity [641]. One major causative factor in tissue inflammation is the uncontrolled overproduction ROS/reactive nitrogen species. The transcriptional regulator NF-κB is an inflammatory reaction factor of major importance that is extremely sensitive to oxidants [552,566,567,568,569,570]. Enhanced IL-6 production by activated NF-κB has been implicated in many pathophysiological dysfunctions of aging ranging, from Alzheimer’s disease to atherosclerosis [642]. CR exhibits a broad and effective anti-inflammatory effect. It blunts age-triggered increases in COX-2 levels and activity through the modulation of NF-κB and IκB, in which COX-2-derived ROS generation decreases. In addition, the production of iNOS, IL-β, IL-6, TNFα, and prostanoids such as thromboxane A2 (TXA2), prostacyclin 2, and prostaglandin E2 is suppressed [14,531]. The prevention of the age-related decline triggered by CR correlates with dampening the reduction of PPAR expression and activity seen during aging. Therefore, under CR conditions, higher PPAR expression may play a role in the suppression of the age-induced increase in inflammation [140]. PPARs are implicated in inflammation at the transcriptional level by interfering with pro-inflammatory mediators such as NF-κB, STAT-1, and activating protein-1, leading to the downregulation of the gene targets of these factors [643,644,645,646]. In this way, PPARα and PPARγ inhibit the expression of inflammatory genes, such as COX-2, iNOS, cytokines, metalloproteases, and acute-phase proteins [549,644]. Inflammatory eicosanoids serve as ligands for PPARs, and the levels of these signaling molecules, including prostaglandins and leukotrienes, increase with age [647].

Each of the three PPAR isotypes exhibits a set of individual anti-inflammatory properties [58]. The anti-inflammatory activity of PPARα is in a great part the result of its interaction with NF-κB. The deletion of PPARα results in a premature and enhanced age-dependent increase in oxidative stress and NF-κB activity [137]. Similarly, aged PPARα KO mice display higher oxidative stress at a younger age and an exacerbated inflammatory response to LPS stimulation [137,648]. In contrast, the administration of PPARα agonists to aged wild-type mice restores the cellular redox balance, as attested by the elimination of constitutively active NF-κB and a loss in spontaneous inflammatory cytokine production [137,649]. The interaction of NF-κB and PPARα is intriguing, because high doses of the PPARα ligands activate NF-κB, whereas low or therapeutic doses of the ligands cause decreased NF-κB activation accompanied by reduced IL-6 production and lipid peroxidation [137]. During CR, PPARα is required, at least partially, to mediate the downregulation of acute-phase genes (C4bp, C9, Mbl1, Orm1, Saa4) that are responsive to inflammatory cytokines [141].

PPARβ/δ also shows anti-inflammatory properties and can suppress, in a ligand-independent manner, inflammatory bowel disease by the dampening of inflammatory signaling [650]. In cultured cardiomyocytes, the PPARβ/δ agonist GW0742 inhibits LPS-induced TNFα secretion, whereas the absence of PPARβ/δ exaggerates LPS-induced TNFα production [651]. The intracerebroventricular administration of high-affinity PPARβ/δ agonists significantly decreases the infarct volume at 24 h of reperfusion after cerebral ischemia in rats, again underscoring the anti-inflammatory and neuroprotective properties of PPARβ/δ [652]. Lastly, the activation of PPARβ/δ by GW0742 protects skeletal muscle against metabolic disorders caused by chronic exposure to a high concentration of sugars by affecting the insulin and inflammatory cascades, including reversal of the diet-induced activation of NF-κB and the expression of both iNOS and intercellular adhesion molecule 1 [653].

PPARγ is undeniably one of the most important and best documented anti-inflammatory factors. PPARγ agonists mitigate inflammatory bowel disease symptoms, reduce inflammation, and are effective in multiple models of ulcerative colitis as well as in Crohn’s disease [654,655,656,657,658,659,660,661,662,663,664,665,666,667]. Functionally, the binding of PPARγ to a DNA-bound repressor complex in macrophages blocks the expression of inflammatory genes by preventing the 19S proteasome-mediated degradation of the repressor complex [668]. Accordingly, the ligands of PPARγ inhibit macrophage activation, stimulate macrophage differentiation into non-inflammatory type M2, and suppress the production of inflammatory cytokines in macrophages and dendritic cells, resulting in increased susceptibility to infection in PPARγ-deletion mouse models [549,644,669,670,671,672]. Of interest, the Pro(12)Ala substitution in PPARγ (rs1801282 C>G), which results in a modest decrease in its transcriptional activity and adipogenic potential, mediates anti-inflammatory benefits. The Pro(12)Ala substitution is associated with a 10-year delay in the onset of multiple sclerosis [673] and with a decreased risk for T2D [442]. Males carrying the 12Ala allele and having coronary artery disease show less widespread atherosclerosis and are protected against 10-year vascular morbidity and mortality [674]. Furthermore, another PPARγ polymorphism (rs 1801282 C>G, rs3856806 C>T) is associated with colorectal cancer risk [675,676]. Mice deficient in colonic PPARγ display more acute infectious colitis [663] and are resistant to conjugated linoleic acid therapy for colitis [677]. The molecular mechanism behind the anti-inflammatory activities of PPARγ includes inhibition of the expression of inflammatory genes encoding cytokines, metalloproteases, and acute-phase proteins, and the regulation of multiple signaling pathways, such as those related to p53 [678], Bcl2 [89], c-Myc, [679], Cox-2 [91,680,681,682], iNOS [683], and Apc/β-catenin [684,685]. Most importantly, PPARγ inhibits NF-κB and NF-κB-driven transcription [89,682]. PPARγ may reduce NF-κB activities in various ways (see the section on PPARs and oxidative stress). Therefore, it is likely that PPARs mediate, at least in part, the anti-inflammatory properties of CR.

### 7.4. Metabolic Adaptation

The shortage of energy during CR leads to a sequence of metabolic changes. Following the depletion of dietary glucose, glycogen is mobilized as an energy supply, and upon prolonged CR, hepatic metabolism shifts to gluconeogenesis to prevent hypoglycemia. Some enzymes connected with hepatic glycolysis, gluconeogenesis, and glycogen metabolism are under the control of PPARα. During fasting, PPARα stimulates glucose import, glycolysis, and glycogenolysis [686,687,688,689]. Accordingly, the expression of several genes involved in gluconeogenesis and glycogen metabolism is reduced in PPARα KO mice [368], and these animals show impaired gluconeogenesis regulation and marked hypoglycemia during fasting [54,55].

Upon prolonged energy restriction, carbohydrate depletion triggers a shift to fat recruitment and ketone body production. This switch between energy sources relies on PPARs. Exercise-elicited glycogen depletion activates PPARβ/δ in rat muscle [690]. We speculate that a similar regulation takes place in fasting-related carbohydrate shortage, which would contribute to PPARβ/δ-driven FA oxidation in muscles. Similarly, the upregulation of the expression of PPARα by CR has been suggested to act as a direct stimulus to enhance FA β-oxidation in the heart [139].

PPARs also regulate the expression of many genes involved in insulin signaling, glucose uptake, lipid metabolism, and ketogenesis, which are affected by CR. Particularly, the metabolism of lipids and ketone bodies in the liver employs PPARα to regulate the expression of most of the rate-limiting enzymes of β-oxidation including ACOX1 (acyl-CoA Oxidase 1), EHHADH (enoyl-CoA hydratase and 3-hydroxyacyl CoA dehydrogenase), carnitine palmitoyltransferases I and II, MCAD (medium-chain acyl-CoA dehydrogenase), LCAD (long chain acyl-CoA dehydrogenase), VLCAD (very long chain acyl-CoA dehydrogenase), and fibroblast growth factor 21, and of ketogenesis, such as HMG-CoA synthase [141,691,692,693,694,695,696,697,698,699,700].

During fasting, PPARα promotes cellular FA uptake and β-oxidation and mediates the adaptation to FA catabolism, lipogenesis, and ketone body synthesis in response to energy depletion [53,54,55]. Consequently, fasting-induced hepatic responses, including elevated FA oxidation and ketogenesis, are all impaired in PPARα-null mice, resulting in hypoketogenesis and liver steatosis [53,54,55]. Similarly, in aged mammals, including humans, the capacity for FA oxidation and hepatic ketogenesis decreases, resulting in reduced energy metabolism as well as increased dyslipidemia [223,224,225]. In healthy men, the L162V substitution of PPARα is associated with higher fasting total cholesterol, low-density lipoprotein cholesterol, and apoB, but not with postprandial parameters [50]. Both PPARα and PPARβ/δ are essential regulators of FA oxidation, and their roles in this process overlap. Of importance, the two PPARs show a distinct primary area of activity, with PPAR*α* activating FA oxidation mainly in the liver and BAT, whereas PPARβ/δ controls lipid metabolism in the pancreas, heart, and skeletal muscle. PPARα does not seem to be involved in the metabolic adaptation of the liver to every-other-day fasting [701].

Reduced energy intake accompanied by increased mobilization of the fat reservoir results in weight loss. The release of energy from WAT involves the inhibition of expression of the lipid-storing PPARγ. Collectively, all three PPARs act as metabolic sensors and play essential roles in lipid and FA metabolism. However, PPARγ is more responsible for fat storage and PPARα and PPARβ/δ are more responsible for energy expenditure. Likely for that reason, a high-fat diet increases the expression of PPARγ in the liver, whereas intermittent fasting decreases it [702]. Genetic variation in the *Pparγ* gene and its target gene *Acsl5* determine the capacity for weight loss under CR [703], and six *Pparγ* single nucleotide polymorphisms are significantly associated with weight reduction in response to CR [704]. Most of the data concerning *Pparγ* polymorphisms focus on the Pro(12)Ala substitution. Based on a report of a population of children in Mexico, Pro(12)Pro homozygosity is the more represented, followed by Pro(12)Ala heterozygosity, and more rarely Ala(12)Ala homozygosity (73.9%:24.5%:1.6%) [705]. The (12)Ala PPARγ protein shows a decreased binding affinity for PPRE and consequently is a weaker stimulator of target gene expression [441,706]. The presence of (12)Ala PPARγ and resistance to CR-induced weight loss were associated in a comparison of women with obesity losing the most weight to those losing the least after 6 weeks of a 900 kcal/day CR [703]. Moreover, PPARγ polymorphism is associated with changes in body mass index (BMI) in response to the total fat intake [707,708], FA composition in the diet [709], and plasma TG response to ω3 FA [710]. This polymorphism also influences weight regain following CR, with women homozygous for Ala(12)Ala gaining more weight compared to women with Pro(12)Pro homozygosity [711], likely indicating lesser metabolic flexibility for Ala(12)Ala individuals.

Long-term CR leads to energy-saving adaptations that can result in a lower resting metabolic rate and decreased body temperature [712,713,714], which is possibly because of reduced thyroid hormone levels. Bezafibrate, a panagonist for all three PPARs, has been reported to induce WAT beiging and thus shows potential for regulating body temperature [715]. Similarly, the activation of PPARα in WAT and BAT results in increased UCP-1 expression and consequently elevated energy dissipation and higher body temperature [716]. Crosstalk between thyroid hormone receptors and PPARs appears to be important for regulating thermogenesis and metabolism [717,718]. In summary, the involvement of PPARs in the metabolic feeding-to-fasting adaptation places these receptors at the center of the proper body response to CR.

### 7.5. Physical Exercise

Exercise, similar to CR, yields multiple beneficial effects. Research outcomes point toward the effectiveness of regular moderate exercise in preventing and delaying several metabolic disorders, chronic diseases, and premature death. Increased physical activity reduces mortality risk from many age-related diseases, including cardiovascular disease, stroke, T2D, certain cancers, hypertension, obesity, depression, and osteoporosis [719,720,721,722,723]. However, in rodents, exercise improves the mean lifespan without increasing maximum longevity [724,725]. Similarly, high physical activity fails to extend maximum lifespan in humans [726]. Compared to exercise, long-term CR in humans improves several biomarkers related to aging [727,728]. Accordingly, exercise has been deemed as unable to fully mimic the beneficial hormonal and/or metabolic changes associated with CR [729]. Therefore, despite a mutual influence with CR on similar molecular pathways and providing multiple advantages, physical activity has been recognized as yielding inferior benefits compared to CR.

Physical activity results in the release of stored energy and elevated levels of FAs, increasing the availability of ligands for PPARs [730]. In rat liver, exercise increases PPARα expression and transcription, and PPARα may mediate the impact of exercise on plasma glucose, TG, and cholesterol [731]. Similarly, PPARα may be involved in the protective effects of exercise against myocardial infarction and for cardiac function by changing the expression of metabolic and inflammatory response regulators and by reducing myocardial apoptosis [732]. However, among PPARs, PPARβ/δ is particularly well-known for its impact on physical performance. First, PPARβ/δ controls muscle development and the adaptive response to exercise, and its overexpression results in a switch to type I muscle fiber [620]. Second, exercise can increase PPARβ/δ expression in skeletal muscle, and this activation is essential for increasing the number of exercise-induced muscle mitochondria [617]. As previously mentioned (see the AMPK and PPARβ/δ section), AMPK and PPARβ/δ are exercise mimetics [322], and their stimulation significantly increases running parameters and promotes muscle remodeling [321,322]. During exercise, the depletion of carbohydrates in skeletal muscle limits endurance. PPARβ/δ represses glycolytic genes in muscle to slow glucose catabolism, reducing the use of carbohydrates at the period near exhaustion. In parallel, PPARβ/δ induces a shift to FAs as the main energy source and thus extends the possible exercise time [623,733]. Consequently, transgenic mice overexpressing muscle-specific PPARβ/δ show enhanced exercise performance, but PPARα-overexpressing animals do not [80,310,620].

Exercise is associated with increased PPARγ DNA-binding activity and expression of its target genes in leukocytes [734]. Similarly, in skeletal muscle and subcutaneous WAT, PPARγ and PGC-1α mRNA expression increases in response to physical training, and these expression changes are proposed to mediate the effect of exercise on insulin sensitivity [735]. Furthermore, the beneficial outcome of low-intensity exercise on plasma lipid levels is exerted via PPARγ [734], and PPARγ1 promotes exercise-induced lipoprotein lipase expression [736]. In addition, the Pro(12)Ala substitution in PPARγ polymorphism is associated with reduced glucose and insulin levels as well as body weight loss in response to exercise [737,738,739,740]. Therefore, as in the case of CR, PPARs play an active role in upstream molecular signaling and beneficial outcomes of exercise.

### 7.6. Hunger

Food withdrawal or limitation inevitably results in a hunger sensation. The physiology of hunger involves a complex network of sensors, hormones, and neuronal signaling. Hunger signaling relies on PPARs, particularly PPARα. For example, allele “A” in *PPARA* rs4253747 (a single nucleotide polymorphism in an intron region) in young men of Han Chinese ancestry is significantly associated with an increased risk for appetite loss at high altitude. In contrast, the “AC” haplotype of *PPARA* rs7292407-rs6520015 in the same cohort had a protective role for high altitude appetite loss [741]. If PPARα is implicated in appetite control, some of its natural ligands should affect hunger sensation. Oleoylethanolamide (OEA), an endogenous ligand of PPARα, is one such compound. This endocannabinoid is produced by enterocytes in response to fat consumption [742], and bile acids modulate its biosynthesis, which requires sympathetic innervation [743,744]. The administration of OEA has an anorectic effect by acting peripherally, reducing meal size or prolonging eating latency, leading to body weight loss [742,745,746,747]. PPARα activation in the proximal small intestine mediates this effect [742,748,749]. The intraperitoneal administration of OEA acutely decreases energy expenditure, as well as ambulatory and spontaneous locomotor activity [750]. Via PPARα, OEA stimulates lipolysis and decreases the neutral lipid content in hepatocytes, as well as serum cholesterol and TG levels, and thereby regulates lipid metabolism [751]. OEA engages afferent sensory fibers of the vagal nerve in the intestine, leading to an increased expression of proto-oncogene c-fos in the nucleus solitary tract and the paraventricular nucleus of the brainstem and hypothalamus, respectively [751], which promotes oxytocin secretion and satiety [752]. Since enterocytes in the small intestine are the first cells to respond to dietary fat intake by increasing OEA production, OEA has been suggested to serve as a gut-derived satiety factor [742].

The other PPARs may also indirectly affect appetite. As a master regulator of adipogenesis, PPARγ plays a crucial role in regulating food intake because WAT secretes a number of endocrine and paracrine satiety mediators, including leptin, adiponectin, and resistin [753]. Inflammation, which is well established as being under PPAR influence, reduces appetite [754]. Particularly, NF-kB, which interacts with all PPARs, has been implicated in appetite suppression [755,756].

### 7.7. Longevity and Aging

According to the “rate-of-living” theory, lifespan differs between species and it correlates with energy metabolic rate and, in general, with body size [757,758]. Thereupon, the level of CR needed to prolong life by a certain percentage varies from species to species. In fact, CR increases lifespan to different extents with a stronger impact on short-living animals, such as *C. elegans* (up to 150%) [759], *Drosophila* [760], or rodents (up to 50%) [761] compared to long-lived species including lemurs [762] and rhesus monkeys [6,763]. Therefore, it is not possible to extrapolate the results obtained for one species to others. Importantly, the level of restriction (10–50%) applied in various studies strongly impacts the outcome [11,12]. Although it is proven that CR causes beneficial metabolic modifications in women and men, the exact amount of calorie intake that is necessary to reach maximum longevity and sustain good health is not known, but it is likely different from person to person. It is also obvious that excessive CR leads to malnutrition with adverse health effects.

Both genetic and environmental factors control the progression of aging. Aging is associated with immunosenescence, increased oxidative stress, decreased hormonal secretion, changes in metabolic rate, mitochondrial function, insulin resistance, and dysregulated lipid metabolism [764,765,766]. The preservation of insulin sensitivity by reducing levels of blood glucose and insulin without compromising glucose fuel may prevent age-related metabolic phenotypes [353]. Glucose metabolism maintenance is a key feature of the anti-aging actions of CR [3]. In fact, genes connected with the insulin/IGF-1 signaling pathway have been proposed as longevity candidate markers [356,357,767]. Paradoxically, impaired insulin signaling through the insulin receptor or its substrates increases rather than decreases lifespan in a number of mouse models [351,356,357]. With regard to insulin sensitivity, the undeniable role of PPARα and PPARβ/δ has already been described above. Worth highlighting here in the context of PPARγ is the role of WAT in proper glucose homeostasis, as attested by the association of lipodystrophy with severe insulin resistance [396].

Aging is accompanied by increased body fat, particularly in the visceral areas as well as in liver and skeletal muscle, which initiates the development of age-associated hypertension, atherosclerosis, inflammatory diseases, dyslipidemia, and T2D [764,765,766]. In fact, WAT, which has also been directly associated with lifespan, is a central physiological component of energy metabolism [351,396], and its development and function rely on PPARγ [106,768,769]. In addition to the direct regulation of adipogenesis from fat-produced PPARγ, intestinal PPARγ regulates body adiposity by sympathetic nervous system signaling in mice on CR [101]. Therefore, through the coordination of glucose homeostasis and adipogenesis, PPARγ might affect longevity [770].

A low expression of PPARγ reduces the lifespan in both lipodystrophic PPARγ1/2-hypomorphic and PPARγ2-deficient mice [771]. We suggest that the reduced fat mass observed during CR is not what results in longevity, but rather that the key factor is proper adipose tissue functionality, including insulin-sensitizing effects. Correspondingly, the human genetic variant genotypes Pro(12)Ala and Ala(12)Ala of PPARγ are associated with leanness, improved insulin sensitivity, and increased lifespan in both humans and mice [440,441,442]. Moreover, gene network analysis has identified PPARγ as one of the “longevity genes” in mouse WAT [771]. However, reports are contradictory concerning the expression of PPARγ in WAT in response to CR. One group found that both 8 weeks of 50% CR and intermittent fasting downregulated PPARγ mRNA and protein expression in the adipose tissue of obese rats [132]. A similar effect was observed in the subcutaneous adipose tissue of obese humans following 10-week CR [772]. In contrast, in intermittently fasting rats, PPARγ2 mRNA levels were approximately two-fold higher than in control or CR animals [773], resulting in a not well-understood effect of restrictive diets on PPARγ2 expression in WAT.

Most of the prominent features of aging are related to PPAR activity, mTOR activity, oxidative stress, inflammation, and metabolism. Moreover, changes in PPAR expression and activity often occur in aging and are reversed by CR [140,224,539,774]. PPAR activity also may be affected indirectly through age-dependent decreases in RXR, the heterodimerization partner of PPARs [775,776,777]. The impact of PPARs can be particularly well observed in mutant models of longevity, such as the dwarf mice. Snell dwarf, Ames dwarf, and “Little” mice display low levels of GH or a defect in GH signaling because of a mutation (generating the GHR-KO strain). All of these dwarf mice are characterized by having a markedly longer lifespan than their wild-type counterparts and share a number of beneficial phenotypic characteristics with rodents on CR diets. Similar to CR animals, dwarf mice are protected from spontaneous and chemically induced cancer, age-dependent declines in immune function, collagen cross-linking, decreased levels of insulin and IGF-1, and increased insulin sensitivity [428]. The increased expression of PPARα and constitutive activation of some of its target genes have been detected in the liver of the dwarf mice [131,778]. The increased expression of genes involved in β- and ω-oxidation of FAs (*Acox1*, *Cyp4a10*, *Cyp4a14*) in the liver of these mice suggests increased FA oxidation, which could be beneficial for insulin sensitivity. PPARα levels are decreased in the muscle of GHR-KO animals, and PPARβ/δ protein levels are downregulated in the liver and skeletal muscle, which mimics the expression profile in wild-type CR mice [136]. The protein levels of PPAR*γ* are elevated in the liver but downregulated in the skeletal muscle of the GHR-KO animals [136]. Furthermore, the overexpression of fibroblast growth factor 21, previously mentioned as a PPARα target gene, extends the lifespan in mice without affecting AMPK or mTOR but blunting GH/IGF-1 signaling in the liver [779].

In contrast to GHR-KO mice, animals overexpressing the bovine GH gene have a markedly shorter lifespan in comparison to their wild-type counterparts. The hepatic expression of PPARα is decreased in these mice, as is the expression of genes involved in FA activation, peroxisomal and mitochondrial β-oxidation, and the production of ketone bodies. Consequently, bovine GH mice exhibit a reduced ability to produce ketone bodies in the fed and fasted states [780]. The antagonistic relationship between PPARs and GH is demonstrated by the fact that the surgical removal of the pituitary gland (hypophysectomization) of rats enhances the expression of PPAR-inducible proteins, which can be reversed by GH infusion [781]. Moreover, STAT5b, a GH-inducible transcription factor, inhibits the ability of PPARα to activate PPARα-dependent reporter gene transcription [782,783], and PPARα downregulates STAT5b [784]. Consequently, PPARs may control lifespan at the level of glucose and lipid metabolism and hormonal regulation.

### 7.8. Microbiota Composition

Microbiota composition changes upon CR have been repeatedly observed [138,785,786,787,788]. CR increases the abundance of bacteria that positively correlate with lifespan, mainly Firmicutes including *Lactobacillus*, Allobaculum, Papillibacter, or Lachnospiraceae. In parallel, CR reduces the occurrence of bacteria that negatively correlate with lifespan, such as Clostridiales, Riminococcaceae, Alistipes, or Rikenella [787,788,789,790,791]. The exact effect of microbiota on the outcome of CR is not fully known, but the microbiota mediates some of the beneficial outcomes of CR, including reduced body weight and decreased blood leptin and insulin levels [791]. We could speculate that there is an effect on metabolism, body fat storage, and the endocrine system of microbiota-driven changes in the production of signaling molecules and ligands for nuclear receptors, including PPARs [101]. Indeed, the interaction of PPARs with the microbiota has been well documented. The expression of PPARα and its target genes coding for rate-limiting enzymes of ketogenesis depends on stimulation by commensal gut microbiota [691,698,792]. Using germ-free mice, we have shown that the microbiota not only promotes harvesting energy from the food but is also generating signals, which regulate the hepatic clock genes and their effector genes such as the PPARs, and several PPARα target genes [793]. Of note, PPARα also mediates signals received from the microbiota via TLRs and contributes to the circadian expression of genes in the intestine and intestinal corticosterone production [794]. Thus, PPARα forwards information from the gastrointestinal flora, which affects host physiology. Furthermore, PPARα has been identified as an important factor in the inflammatory response of the intestine to commensal microbiota [795]. It regulates the expression of IL-22, the antimicrobial peptides Reg3β and Reg3γ, and calprotectin [795]. In the context of restrictive diets, the microbiota mediates the stimulatory effect of intermittent fasting on beige fat development [796]. Similarly, the deletion of PPARα triggers the upregulation of UCP1 expression in WAT [796]. PPARβ/δ, which is constitutively expressed in the intestine at a high level [45] and takes part in the differentiation of intestinal cells, is indirectly involved in the secretion of antimicrobial peptides [74,650,797]. Therefore, it influences gut microbiota composition.

The expression and activity of PPARγ are induced in the gut by multiple nutrients [114], bacterial metabolites, and bacterial by-products [115,116,117,118], and the presence of specific bacterial strains [117,119,120] stimulates PPARγ expression and activity. However, CR has been shown to limit the production of butyrate [787], which is one of the short chain fatty acids (SCFAs) that is known to activate PPARγ [116,117]. Moreover, the microbiota affects the liver circadian rhythm by modulating the activity of PPARγ expressed in the liver [798]. Of note, PPARγ is responsible for the selective killing of bacteria associated with inflammatory bowel disease by stimulating the expression of β-defensins and the maintenance of innate antimicrobial immunity in the colon [799].

Thus, there are reciprocal interactions between PPARs and gut microbiota in which PPARs can be activated by bacteria and regulate the intestinal microbiota composition [800]. The additional impact of CR on the expression PPARs points to a potential role for PPARs in the response of microbiota to CR.

## 8. Conclusions

After remarkable achievements in medical research that have translated into a notable increase in life expectancy, the current focus is more on increasing disease-free years. With the potential to alleviate numerous health conditions while extending the lifespan, CR remains a relevant candidate in health-related research. Therefore, the current recommendation on energy intake should be revised, particularly for individuals with a high risk of developing metabolic, inflammatory, or neurodegenerative diseases.

The beneficial impact of several restrictive approaches including multiple models of intermittent fasting [801,802,803,804] and fasting-mimicking diet [805,806,807] has been proven. These diets imply temporal restriction without long-term energy deprivation or prolonged hunger making them less restrictive than CR, easier to apply in everyday life, more flexible for various lifestyles, and therefore more plausible for a wide population. The majority of available studies concerning intermittent fasting are observational studies focused on weight loss, cardiovascular risk, and inflammation. However, knowledge of the molecular mechanism behind the observed effects is still limited [808,809]. More studies comparing CR and other specific restrictive diets, in terms of molecular pathways and health outcomes, are needed to identify which restrictive approach is more efficient. Eventually, the aim is to encourage the use of such diets as a means to prevent diseases. Currently, the large variety of intermittent fasting and CR protocols increases the complexity of this task. Interestingly, the involvement of PPARs in the impact of the intermittent fasting and fasting-mimicking diet has not yet been verified.

Despite the lack of a fully revealed network of pathways underlying CR, the current review gives a detailed overview of how CR exerts its effects on the whole organism and which of the many outcomes are mediated by PPARs. During CR, the energy and nutrient sensor pathways involving mTOR, AMPK, insulin signaling, and SIRT are tightly interconnected, resulting in reduced oxidative stress and inflammation, increased autophagy, improved mitochondrial function, and the regulation of metabolism, hunger, and microbiota composition. As discussed, multiple connections have been found between PPARs and regulatory responses to CR and fasting. PPARs take part in managing the initial shortage of energy, modulating the main signaling pathways, adjusting metabolism to prolonged low energy intake, and mediating direct and long-term outcomes. Each of these roles could contribute to improvements related to common diseases such as diabetes and cancer, and even neurodegenerative diseases. Therefore, PPAR agonist treatments together with CR has great potential for synergistic effects to be explored in future experimental and clinical studies.

## Figures and Tables

**Figure 1 cells-09-01708-f001:**
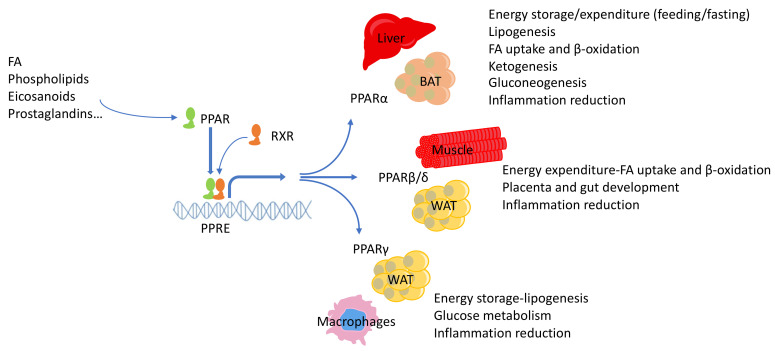
Activation and main functions of peroxisome proliferator-activated receptors (PPARs) in different tissues. PPARs share fatty acids (FA) as common ligands, peroxisome proliferator response elements (PPRE) as their DNA binding site, and retinoid X receptors (RXR) as their heterodimer partner. However, each PPAR shows distinct expression and function patterns. The dominant role of PPARα is connected to metabolic adjustment in the liver and brown adipose tissue (BAT). PPARβ/δ is primarily associated with muscle and white adipose tissue (WAT) metabolism, as well as with organ development. PPARγ is a master regulator of adipogenesis and WAT maintenance and plays an important anti-inflammatory role. However, this cartoon represents a schematic and simplified view of much more complex patterns.

**Figure 2 cells-09-01708-f002:**
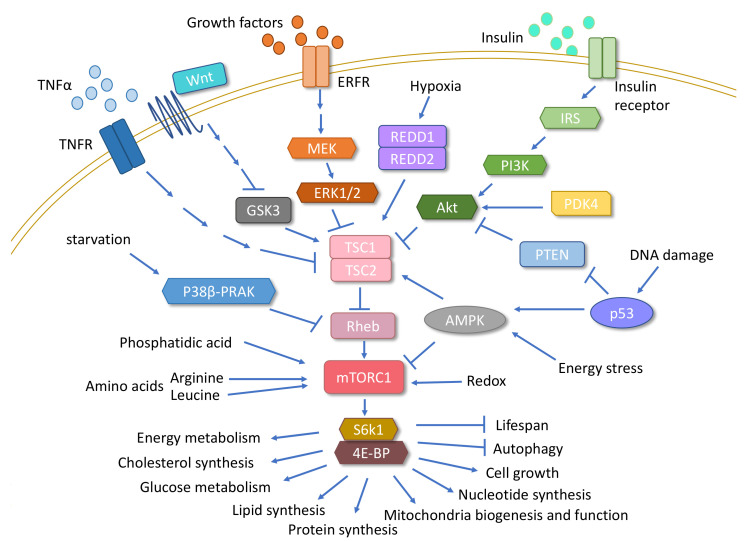
The main signaling pathways associated with mammalian target of rapamycin (mTOR). mTOR integrates inputs from multiple sources including growth factors, insulin, stress, energy balance, oxygen, and nutrients, and it controls many major downstream processes, including metabolism, macromolecule synthesis, mitochondria function, cell growth, and autophagy.

**Figure 3 cells-09-01708-f003:**
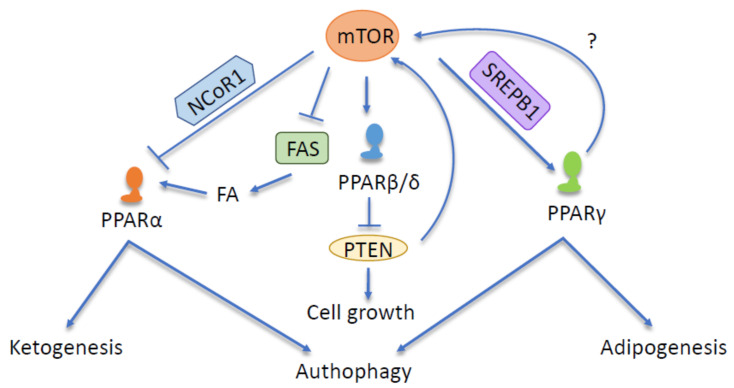
Interactions between PPARs and mTOR. mTOR interacts with all PPARs, resulting in the modulation of ketogenesis, autophagy, and adipogenesis.

**Figure 4 cells-09-01708-f004:**
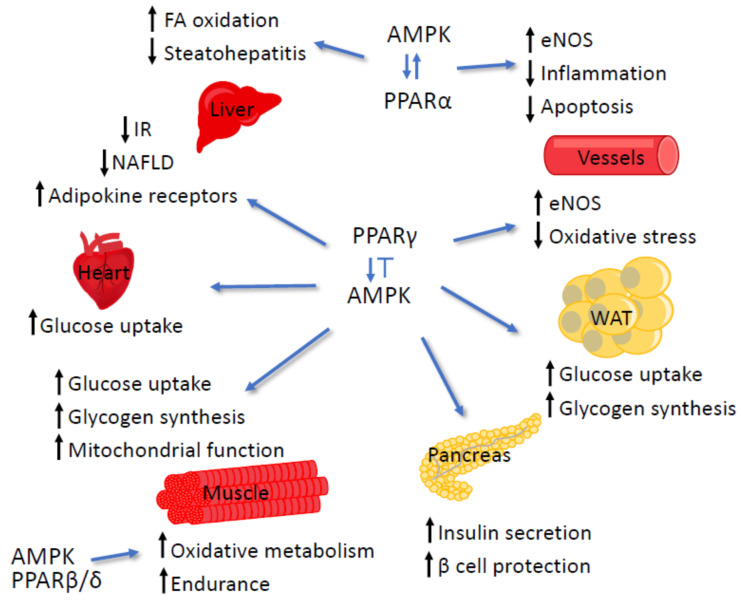
The tissue-specific outcomes of the interaction between PPARs and adenosine monophosphate (AMP)-activated protein kinase (AMPK). PPARγ interacts with AMPK in multiple tissues including blood vessels, WAT, pancreas, muscle, heart, and liver, leading to enhanced metabolism as well as reduced oxidative stress and inflammation. PPARα, in cooperation with AMPK, affects metabolism in the liver as well as reducing inflammation and apoptosis in blood vessels, whereas PPAR β/δ with AMPK affects muscle performance.

**Figure 5 cells-09-01708-f005:**
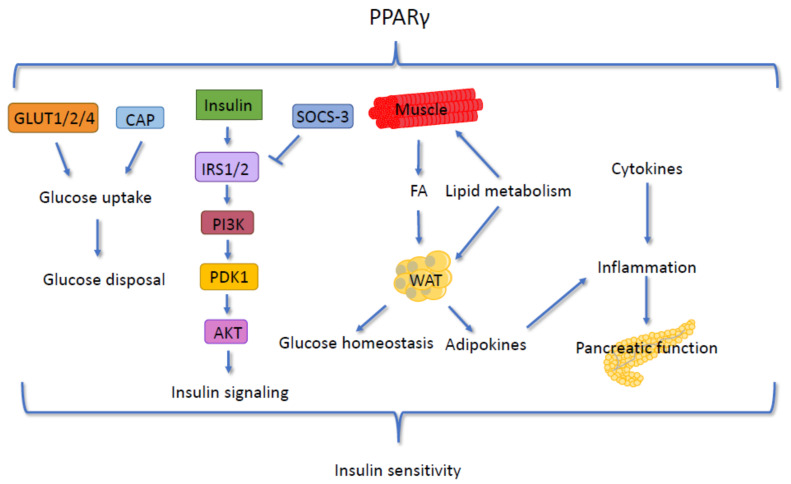
Pathways in which PPARγ activity leads to increased insulin sensitivity. PPARγ affects insulin sensitivity by managing glucose uptake and disposal, enhancing insulin signaling, and maintaining functioning WAT and pancreas.

**Figure 6 cells-09-01708-f006:**
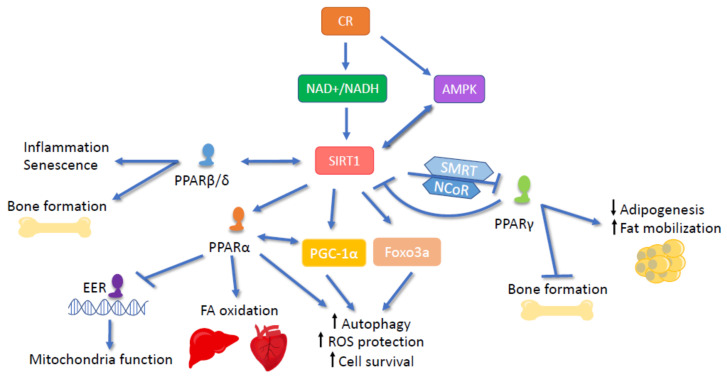
The interaction between sirtuin 1 (SIRT1) and PPARs. Caloric restriction (CR)-triggered energy shortage leads to the activation of SIRT1 and its interaction with PPARs. Each of these interactions results in a distinct outcome.

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
