# Peer review of "Peroxisome Proliferator-Activated Receptors and Caloric Restriction—Common Pathways Affecting Metabolism, Health, and Longevity"

_cells, 2020, doi:10.3390/cells9071708_

Round 1
Reviewer 1 Report
The authors are providing a comprehensive review of the impact of calorie restriction (CR) and signaling pathways related to PPARs, and ultimately the functional outcome of CR. Though already lengthy, I would suggested the following brief additions to further strengthen the manuscript:
pg 14 - while introducing the impact of CR on insulin signaling (a highly researched topic), the authors could have discussed the role of Akt in further depth, particularly as much consideration was given to AMPK
pg 29 - while discussing the impact of CR on longevity, the authors could have additionally discussed potential similarities/differences from basic organisms (ie worms) to more complex (ie primates and humans).
Also, though there were a few references to intermittent fasting, a paragraph discussion of mode of CR including intermittent fasting would be pertinent as this has gained popularity as a potential weight loss method (with various levels of success).
Author Response
We thank the reviewer his/her constructive evaluation of our manuscript and for taking time, in the current difficult sanitary situation, to read a long review and propose ameliorations that are welcome.
Please find our responses to the reviewer’s concerns below. The reviewers’ comments are in black with our specific responses red.
The authors are providing a comprehensive review of the impact of calorie restriction (CR) and signaling pathways related to PPARs, and ultimately the functional outcome of CR. Though already lengthy, I would suggested the following brief additions to further strengthen the manuscript:
pg 14 - while introducing the impact of CR on insulin signaling (a highly researched topic), the authors could have discussed the role of Akt in further depth, particularly as much consideration was given to AMPK
Additional information concerning Akt and its activity in the context of CR has been added on page 14.
pg 29 - while discussing the impact of CR on longevity, the authors could have additionally discussed potential similarities/differences from basic organisms (ie worms) to more complex (ie primates and humans).
Discussion concerning the potential of CR to extend lifespan as well as differences in experimental outcomes for various species has been included in the “Longevity and aging” section.
Also, though there were a few references to intermittent fasting, a paragraph discussion of mode of CR including intermittent fasting would be pertinent as this has gained popularity as a potential weight loss method (with various levels of success).
In the section “Conclusions” an appropriate paragraph discussing intermittent fasting and fasting-mimicking diets has been added.
Reviewer 2 Report
The review article by Duszuka and colleagues summarizes the current understanding between the links between caloric restriction (CR) and PPAR physiology. The review is very thorough (775 references), well written and devoid of typographical and grammatical errors. Figures are appropriate. In a time whereby we have extensive chronic metabolic and neurodegenerative disease prevalence, understanding the physiological effects of CR are important and necessary to advance potential treatment avenues. Therefore, review articles play an important function to disperse known information. I have several comments for the authors.
Title: In my opinion I would recommend dropping “At the Crossroads and Roundabouts of PPARs and” in the title. I do not think it adds anything.
Introduction. It may be helpful to have a bit more background on what CR involves. From a translational point of view what sort of decrease in energy intake or how many calories per day intake are we looking at in humans and what is used in the various animal studies you mention. Does it matter in terms of macronutrient ratio? What is compliance to a CR diet like in humans? This will set the scene a little better for those new to the area.
Line 972/3 ; “mitochondria biogenesis is relatively high in various tissues such as in the brain, heart, liver, and particularly WAT”. WAT is not particularly a location of relatively high mitochondria biogenesis. BAT, skeletal muscle and proximal tubules of kidney are far more inundated with mitochondria and creation of mitochondria.
Line 1330: Microbiota composition: What is the interaction between various SCFA’s and PPAR’s with changing microbiota composition in response to CR? How does the general profile of the microbiota change with CR?
I apologise if I missed your commentary on it, but what about the synergistic effects of PPAR activators or inhibitors in combination with CR or a percentage of a CR diet? In other words, could we look to target PPARs pharmaceutically combined with a partial CR diet for those that cannot commit to full CR? Is this an avenue to optimise effects?
Author Response
We thank the reviewer his/her enthusiasm and constructive evaluation of our manuscript and for taking time, in the current difficult sanitary situation, to read a long review and propose ameliorations that are welcome.
Please find our responses to the reviewer’s concerns below. The reviewers’ comments are in black with our specific responses red.
The review article by Duszuka and colleagues summarizes the current understanding between the links between caloric restriction (CR) and PPAR physiology. The review is very thorough (775 references), well written and devoid of typographical and grammatical errors. Figures are appropriate. In a time whereby we have extensive chronic metabolic and neurodegenerative disease prevalence, understanding the physiological effects of CR are important and necessary to advance potential treatment avenues. Therefore, review articles play an important function to disperse known information. I have several comments for the authors.
Title: In my opinion I would recommend dropping “At the Crossroads and Roundabouts of PPARs and” in the title. I do not think it adds anything.
Thank you for your comment. The manuscript’s title has been shortened accordingly.
Introduction. It may be helpful to have a bit more background on what CR involves. From a translational point of view what sort of decrease in energy intake or how many calories per day intake are we looking at in humans and what is used in the various animal studies you mention. Does it matter in terms of macronutrient ratio? What is compliance to a CR diet like in humans? This will set the scene a little better for those new to the area.
The required information was added to the "Introduction" section. However, data on the compliance to CR in humans vary considerably in different publications. Compliance is influenced by multiple factors. Therefore, it was not possible to conclude with a short answer. To avoid making the review even longer, we did not include this information. We hope that the reviewer can agree with us.
Line 972/3 ; “mitochondria biogenesis is relatively high in various tissues such as in the brain, heart, liver, and particularly WAT”. WAT is not particularly a location of relatively high mitochondria biogenesis. BAT, skeletal muscle and proximal tubules of kidney are far more inundated with mitochondria and creation of mitochondria.
Thank you. The mistake has been corrected. We changed WAT to BAT as should have been from the beginning.
Line 1330: Microbiota composition: What is the interaction between various SCFA’s and PPAR’s with changing microbiota composition in response to CR? How does the general profile of the microbiota change with CR?
So far, the only SCFA identified to interact with PPARs is butyrate and studies show that CR limits butyrate production. This information together with the characterization of microbiota composition changes during CR have been introduced in the “Microbiota composition” section.
I apologise if I missed your commentary on it, but what about the synergistic effects of PPAR activators or inhibitors in combination with CR or a percentage of a CR diet? In other words, could we look to target PPARs pharmaceutically combined with a partial CR diet for those that cannot commit to full CR? Is this an avenue to optimise effects?
The effects of PPARs activators or inhibitors in combination with CR have never been investigated before. Hence, we cannot discuss any relevant results. However, it is a very interesting point to stimulate future studies. Therefore, we enriched the "Conclusion" section based on your suggestion.